# Differentially Private Deep Model-Based Reinforcement Learning

## Abstract

We address private deep offline reinforcement learning (RL), where the goal is to train a policy on standard control tasks that is differentially private (DP) with respect to individual trajectories in the dataset. To achieve this, we introduce PRIMORL, a model-based RL algorithm with formal differential privacy guarantees. PRIMORL first learns an ensemble of trajectory-level DP models of the environment from offline data. It then optimizes a policy on the penalized private model, without any further interaction with the system or access to the dataset. In addition to offering strong theoretical foundations, we demonstrate empirically that PRIMORL enables the training of private RL agents on offline continuous control tasks with deep function approximations, whereas current methods are limited to simpler tabular and linear Markov Decision Processes (MDPs). We furthermore outline the trade-offs involved in achieving privacy in this setting.

## 1 Introduction

Despite Reinforcement Learning's (RL) notable advancements in various tasks, there have been many obstacles to its adoption for the control of real systems in the industry. In particular, online interaction with the system may be impractical or hazardous in real-world scenarios. Offline RL (Levine et al., 2020) refers to the set of methods enabling the training of control agents from static datasets. While this paradigm shows promise for real-world applications, its deployment is not without concerns. Many studies have warned of the risk of privacy leakage when deploying machine learning models, as these models can memorize part of the training data. For instance, Rigaki & Garcia (2020) review the proliferation of sophisticated privacy attacks. Of the various attack types, membership inference attacks (MIAs) (Shokri et al., 2017) stand out as the most prevalent. In these attacks, the adversary, with access to a black-box model trainer, attempts to predict whether a specific data point was part of the model's training data. Unfortunately, RL is no exception to these threats. In recent work, Gomrokchi et al. (2023) exploit the temporal correlation of RL samples to perform powerful membership inference attacks using convolutional neural classifiers. More precisely, they demonstrate that given access to the output policy, an adversary can learn to infer the presence of a specific trajectory — which is the result of a sequence of interactions between a user and the system — in the training dataset with great accuracy. The threat of powerful MIAs is particularly concerning in reinforcement learning, where a trajectory can unveil sensitive user information. For instance, when using RL to train autonomous vehicles (Kiran et al., 2022), we need to collect a large number of trips that may disclose locations and driving habits. Similarly, a browsing journey collected to train a personalized recommendation engine may contain sensitive information about the user's behavior (Zheng et al., 2018). In healthcare, RL's potential for personalized treatment recommendation (Liu et al., 2022) underscores the need to safeguard patients' treatment and health history.

A large body of work has focused on protecting against privacy leakages in machine learning. Differential Privacy (DP), which allows learning models without exposing sensitive information about any particular user in the training dataset, has emerged as the gold standard. While successfully applied in various domains, such as neural network training (Abadi et al., 2016) and multi-armed bandits (Tossou & Dimitrakakis, 2016), extending differential privacy to reinforcement learning poses challenges. In particular, the many ways of collecting data and the correlated nature of training samples resulting from online interactions make it difficult to come up with a universal and meaningful DP definition in this setting, despite several attempts such as local and joint DP. In addition to its practical significance, the offline RL setting arguably offers a more natural framework for privacy

compared to the classic online setting. An offline RL method can indeed be seen as a black-box randomized algorithm $h$ taking in as input a fixed dataset $\mathcal{D}$, partitioned in trajectories, and outputting a policy $\hat{\pi}$. An adversary with access to $\hat{\pi}$ may successfully learn to infer the membership of a specific trajectory in $\mathcal{D}$, which can, as emphasized before, reveal sensitive user information. Hence, similarly to Qiao & Wang (2023a), we use the following informal DP definition for offline RL, which we refer to as *trajectory-level differential privacy* (TDP): adding or removing a single trajectory from the input dataset of an offline RL algorithm must not impact significantly the distribution of the output policy.

While reinforcement learning encounters the same privacy challenges as other areas of machine learning, no existing work has proposed a private RL method that matches the versatility, scalability, and empirical effectiveness of DP-SGD (Abadi et al., 2016) for supervised learning. Indeed, existing research largely remains theoretical and demonstrates limited practical applicability. In the online setting, numerous private algorithms have been developed (*e.g.*, Vietri et al. (2020), Garcelon et al. (2021), Qiao & Wang (2023b)) but their scope remains restricted to tabular and linear Markov Decision Processes (MDPs) with finite horizon. Qiao & Wang (2023a) have proposed the first private algorithms for offline RL, building on value iteration methods, but they present similar limitations. These approaches cannot intrinsically scale to the problems typically encountered in deep RL, leaving a huge gap between the current private RL literature and real-world applications. This work addresses this gap by introducing the first deep RL method with provable privacy guarantees. In contrast to previous work, our method is applicable to general MDPs with continuous state and action spaces and deals with the classic $\gamma$-discounted setting, paving the way for enhanced applications of private RL in complex, risk-sensitive scenarios.

**Contributions.** While the current differentially private RL literature is mainly theoretical and has limited practical relevance, this work is the first attempt to tackle deep RL problems with differential privacy guarantees. Specifically, we address the offline setting under the well-founded concept of trajectory-level privacy, and introduce a model-based approach named PRIMORL. Protecting entire trajectories, rather than individual examples, precludes the use of vanilla optimizers such as DP-SGD. Additionally, the standard approach of using bootstrap ensembles to handle model uncertainty presents an extra challenge in the private setting, as the ensemble size directly impacts the privacy budget. A key contribution of this work is therefore the introduction of a training method for model ensembles that ensures differential privacy at the trajectory level and effectively controls the privacy budget. We also provide a theoretical analysis of how private training influences model reliability. We then perform policy optimization under the resulting pessimistic private model and prove the formal privacy guarantees of the resulting policy. We show empirically that PRIMORL can train private policies with competitive privacy-performance trade-offs on standard continuous control benchmarks, demonstrating the potential of our approach.

## 2 RELATED WORK

Offline RL (Levine et al., 2020; Prudencio et al., 2022) focuses on training agents without further interactions with the system, making it essential in scenarios where data collection is impractical (Singh et al., 2022; Liu et al., 2020; Kiran et al., 2022). Model-based RL (Moerland et al., 2023) can further reduce costs or safety risks by using a learned environment model to simulate beyond the collected data and improve sample efficiency (Chua et al., 2018). Argenson & Dulac-Arnold (2021) demonstrate that model-based offline planning, where the model is trained on a static dataset, performs well in robotic tasks. However, offline RL faces challenges like *distribution shift* (Fujimoto et al., 2019), where the limited coverage of the dataset can lead to inaccuracies in unexplored state-action regions, affecting performance. Methods like MOPO (Yu et al., 2020), MOREL (Kidambi et al., 2020), and COUNT-MORL (Kim & Oh, 2023) address this by penalizing rewards based on model uncertainty, achieving strong results on offline benchmarks. Still, key design choices in offline MBRL require further exploration, as highlighted by Lu et al. (2022).

On the other hand, Differential Privacy (DP), established by Dwork (2006), has become the standard for privacy protection. Recent research has focused on improving the privacy-utility trade-off, with relaxations of DP and advanced composition tools enabling tighter privacy analyses (Dwork et al., 2010; Dwork & Rothblum, 2016; Bun & Steinke, 2016; Mironov, 2017a). Notably, DP-SGD (Abadi et al., 2016) has facilitated the development of private deep learning algorithms, despite ongoing practical challenges (Ponomareva et al., 2023). Concurrently, sophisticated attack strategies have

underscored the necessity for robust DP algorithms (Rigaki & Garcia, 2020). Recent studies have shown that reinforcement learning (RL) is also vulnerable to privacy threats (Pan et al., 2019; Prakash et al., 2022; Gomrokchi et al., 2023). As RL is increasingly applied in personalized services (den Hengst et al., 2020), the need for privacy-preserving training techniques is critical. Although DP has been successfully extended to multi-armed bandits (Tossou & Dimitrakakis, 2016; Basu et al., 2019), existing RL algorithms (*e.g.*, Vietri et al. (2020), Zhou (2022), Qiao & Wang (2023b)) with formal DP guarantees mainly apply to episodic tabular or linear MDPs and lack empirical validation beyond basic simulations. Moreover, private offline RL remains underexplored. Only Qiao & Wang (2023a) have proposed DP offline algorithms, which, while theoretically strong, are also restricted to finite-horizon tabular and linear MDPs. Consequently, no existing work has introduced DP methods that can handle deep RL environments in the infinite-horizon discounted setting, a critical step toward deploying private RL algorithms in real-world applications. With this work, we aim to fill this gap by proposing a differentially private, deep model-based RL method for the offline setting.

## 3 PRELIMINARIES

### 3.1 OFFLINE MODEL-BASED REINFORCEMENT LEARNING

We consider an infinite-horizon discounted MDP, that is a tuple $\mathcal{M} = (\mathcal{S}, \mathcal{A}, P, r, \gamma, \rho_0)$ where $\mathcal{S}$ and $\mathcal{A}$ are respectively the state and action spaces, $P : \mathcal{S} \times \mathcal{A} \longrightarrow \Delta(\mathcal{S})$ the transition dynamics (where $\Delta(\mathcal{X})$ denotes the space of probability distributions over $\mathcal{X}$), $r : \mathcal{S} \times \mathcal{A} \longrightarrow [0, 1]$ the reward function, $\gamma \in [0, 1)$ a discount factor and $\rho_0 \in \Delta(\mathcal{S})$ the initial state distribution. The dynamics satisfy the Markov property, *i.e.*, the next state $s'$ only depends on current state and action. The goal is to learn a policy $\pi : \mathcal{S} \longrightarrow \Delta(\mathcal{A})$ maximizing the expected discounted return $\eta_{\mathcal{M}}(\pi) := \mathbb{E}_{\tau \sim \pi, \mathcal{M}}[R(\tau)]$, where $R(\tau) = \sum_{t=0}^{\infty} \gamma^t r_t$. The expectation is taken w.r.t. the trajectories $\tau = ((s_t, a_t, r_t))_{t \geq 0}$ generated by $\pi$ in the MDP $\mathcal{M}$, *i.e.*, $s_0 \sim \rho_0$, $s_{t+1} \sim P(\cdot|s_t, a_t)$ and $a_t \sim \pi(\cdot|s_t)$.

In offline RL, we assume access to a dataset of $K$ trajectories $\mathcal{D}_K = (\tau_k)_{k=1}^{K}$, where each $\tau_k = (s_t^{(k)}, a_t^{(k)}, r_t^{(k)})_{t \geq 0}$ has been collected with an unknown behavioral policy $\pi^B$. $\tau_k$ can be seen as the result of the interaction of a user $u_k$ with the environment. The objective is then to learn a policy $\hat{\pi}$ from $\mathcal{D}_K$ (without any further interaction with the environment) which performs as best as possible in $\mathcal{M}$. To achieve this goal, we consider a model-based approach. In this context, we learn estimates of both the transition dynamics and the reward function, denoted $\hat{P}$ and $\hat{r}$ respectively, from the offline dataset $\mathcal{D}_K$. This results in an estimate of the MDP $\hat{\mathcal{M}} = (\mathcal{S}, \mathcal{A}, \hat{P}, \hat{r}, \gamma, \rho_0)$. We can then use the model $\hat{\mathcal{M}}$ as a simulator of the environment to learn a policy $\hat{\pi}_{\hat{\mathcal{M}}}$, without further access to the dataset or interactions with the real environment modeled by $\mathcal{M}$. Note that if the policy $\hat{\pi}_{\hat{\mathcal{M}}}$ is trained to maximize the expected discounted return in the MDP model $\hat{\mathcal{M}}$, *i.e.*, $\hat{\pi}_{\hat{\mathcal{M}}} \in \text{argmax}\, \eta_{\hat{\mathcal{M}}}(\pi)$, we eventually want to evaluate the policy in the true environment $\mathcal{M}$, that is using $\eta_{\mathcal{M}}$.

### 3.2 DIFFERENTIAL PRIVACY

When learning patterns from a dataset, differential privacy (Dwork, 2006) protects against the leakage of sensitive information in the data by ensuring that the output of the algorithm does not change significantly when adding or removing a data point, as formally stated in Definition 3.1.

**Definition 3.1.** $(\epsilon, \delta)$-*differential privacy.* Given $\epsilon > 0$, $\delta \in [0, 1)$, a *mechanism* $h$ (*i.e.*, a randomized function of the data) is $(\epsilon, \delta)$-DP if for any pair of datasets $D, D'$ that differ in at most one element (referred to as *neighboring datasets*, and denoted $d(D, D') = 1$), and any subset $\mathcal{E}$ in $h$'s range:

$$\mathbb{P}(h(D) \in \mathcal{E}) \leq e^{\epsilon} \cdot \mathbb{P}(h(D') \in \mathcal{E}) + \delta .$$

In particular, $\epsilon$ controls the strength of the privacy guarantees, decreasing as $\epsilon$ grows. To achieve $(\epsilon, \delta)$-DP, the standard approach is to add a zero-mean random noise to the output of the (non-private) function $f$, whose magnitude $\sigma$ scales with $\Delta_{\ell}(f)/\epsilon$, where $\Delta_{\ell}(f) := \max_{d(D,D')=1} \|f(D) - f(D')\|_{\ell}$ is the sensitivity of $f$. One of the most used DP mechanisms is the *Gaussian mechanism*, which provably guarantees $(\epsilon, \delta)$-DP for $\epsilon, \delta \in (0, 1)$ by adding random noise from a Gaussian distribution with magnitude $\sigma = \epsilon^{-1}\sqrt{2\log(1.25/\delta)} \cdot \Delta_2(f)$. From such simple mechanisms, we can derive

Figure 1: PRIMORL with its two main components: (1) private model training; (2) MBPO.

complex DP algorithms using the *sequential* and *parallel composition* properties of DP, as well as its *immunity to post-processing* (*i.e.*, if $h$ is $(\epsilon, \delta)$-DP and $g$ is data-independent, $g \circ h$ remains $(\epsilon, \delta)$-DP).

The Gaussian mechanism is central to DP-SGD (Abadi et al., 2016), a learning algorithm that modifies classic SGD to ensure (approximate) differential privacy. By adding Gaussian noise to the gradients and bounding their norm by a constant $C$, DP-SGD enables private neural network training (Ponomareva et al., 2023). To track the total privacy budget $\epsilon_{\text{tot}}$ spent by DP-SGD, Abadi et al. (2016) developed the *moments accounting* method that provides a $\left(\mathcal{O}(q\epsilon\sqrt{T}), \delta\right)$-DP guarantee, where $q$ is the sampling ratio, $T$ is the number of iterations, and $\epsilon$ is the privacy parameter. DP-SGD relies strongly on privacy amplification by sub-sampling (Balle et al., 2018). Studies have also analyzed error bounds for DP-SGD under various loss assumptions (Bassily et al., 2014; Kang et al., 2023).

# 4  DIFFERENTIALLY PRIVATE MODEL-BASED OFFLINE REINFORCEMENT LEARNING

We now describe our model-based approach for learning differentially private RL agents from offline data, which we call PRIMORL (for Private Model-Based Offline RL). After defining trajectory-level differential privacy (TDP) in offline RL (Section 4.1), we address the learning of a private model from offline data (Section 4.2). Finally, we demonstrate how we optimize a policy under the private model (Section 4.3). Exploiting the *post-processing* property of DP, we show that ensuring model privacy alone is enough to achieve a private policy. Figure 1 provides a high-level description of PRIMORL.

## 4.1  TRAJECTORY-LEVEL PRIVACY IN OFFLINE REINFORCEMENT LEARNING

In supervised learning, differential privacy is typically applied at the *example-level*, under the assumption that the examples in the dataset are independent. If this assumption is already questionable in the supervised setting, it certainly does not hold in RL where the transitions $(s_t, a_t, r_t)$ are obviously correlated. Several works, for instance Liu et al. (2016), have demonstrated that data correlation degrades privacy guarantees in the traditional *per-example* setting. It thus appears that protecting individual transitions is insufficient in RL, calling instead for data protection at the *trajectory level*.

We introduce the following formal definition for trajectory-level differential privacy (TDP) in offline RL which protects whole trajectories. It states that the learned policy is roughly the same for two offline datasets $D, D'$ where $D'$ is obtained by adding or removing one full trajectory from $D$. It can be seen as a reformulation of the definition used in Qiao & Wang (2023a), which is the first work to tackle differential privacy in this setting.

**Definition 4.1.** $(\epsilon, \delta)$-*TDP*. Let $h$ be an offline RL algorithm, that takes as input an offline dataset and outputs a policy. Given $\epsilon > 0$ and $\delta \in (0, 1)$, $h$ is $(\epsilon, \delta)$-TDP if for any trajectory-neighboring datasets $\mathcal{D}_K, \mathcal{D}_{K \setminus \{k\}}$, and any subset of policies $\Pi$:

$$\mathbb{P}\left(h(\mathcal{D}_K) \in \Pi\right) \leq e^{\epsilon} \cdot \mathbb{P}\left(h(\mathcal{D}_{K \setminus \{k\}}) \in \Pi\right) + \delta \ .$$

### 4.2 Model Learning with Differential Privacy

Following previous work (Yu et al., 2020; Kidambi et al., 2020), we jointly model the transition dynamics $\hat{P}$ and reward $\hat{r}$ with a Gaussian distribution $\hat{M}$ conditioned on the current state and action. Its mean and covariance are parameterized with neural networks $\theta = (\phi, \psi)$:

$$\hat{M}_\theta \left( \Delta_t^{t+1}(s), r_t | s_t, a_t \right) = \mathcal{N} \left( \mu_\phi(s_t, a_t), \Sigma_\psi(s_t, a_t) \right) \ .$$

To carry out uncertainty estimation (see Section 4.3), we train an ensemble of $N$ models $\hat{M}_{\theta_i}$, $i \in [\![1, N]\!]$, all sharing the same architecture. The core aspect of PRIMORL, as illustrated in Figure 1, is therefore to learn a trajectory-level DP dynamics model ensemble.

This poses two major challenges. First, the traditional approach to privatize neural networks, DP-SGD, is designed for example-level privacy and is unsuitable for guaranteeing TDP. Moreover, since the training of all the models in the ensemble consumes the same dataset $\mathcal{D}_K$, we must deal with the dependence of the privacy budget on the ensemble size $N$. A key contribution of our work, developed in Section 4.2.1, is thus to introduce a training method that 1) guarantees privacy at the trajectory level and 2) efficiently manages the privacy budget across an ensemble of models.

#### 4.2.1 Trajectory-level DP Training for Model Ensembles

As DP-SGD ensures per-example privacy by clipping each per-example gradient, limiting the contribution of each data point to the final model, the key to achieving trajectory-level privacy is to compute and clip per-trajectory updates. Therefore, our training method partitions the dataset by trajectories, *i.e.*, $\mathcal{D}_K = \bigcup_{k=1}^{K}\{\tau_k\}$, computes independent updates from each trajectory's data, and bounds the $L_2$-norm of each update before aggregation. This idea has been developed in McMahan et al. (2017) to achieve user-level privacy when training recurrent language models. Building on prior training algorithms from federated learning, they introduce DP-FEDAVG, which leverages privacy amplification by sub-sampling to achieve competitive privacy-utility trade-offs in language modeling. To address the unique privacy challenges of our task, we build on this approach and adapt it to ensembles of dynamics models. We present the resulting training procedure in Algorithm 1.

The core idea behind TDP MODEL ENSEMBLE TRAINING is to draw, at each iteration $t$, a random subset $\mathcal{U}_t$ of the $K$ trajectories (line 2) using Poisson sampling. Each trajectory is drawn with probability $q$, resulting in an expected $qK$ trajectories being selected per step. The sampling ratio $q$ plays a critical role in determining the strength of privacy guarantees. Specifically, a smaller $q$ reduces the likelihood of any given trajectory being included in an update, thereby limiting its influence on the final model — this forms the basis of privacy amplification by sub-sampling. However, in the offline RL setting, where trajectory data is highly correlated, $q$ must remain large enough to ensure that the model update incorporates a sufficiently diverse set of trajectories. Interestingly, while the theoretical analysis of common private deep learning methods like DP-SGD relies on Poisson sampling, most implementations actually use fixed-size batches with shuffling in practice, in order to overcome the computational challenges due to batches of varying size. This can lead to significant underestimation of the actual privacy leakage, as pointed out in Chua et al. (2018). Our implementation, however, does indeed use Poisson sampling, allowing us to compute correct theoretical privacy guarantees.

For each trajectory $\tau_k \in \mathcal{U}_t$, the clipped gradients $\{\Delta_{i,k}^{\text{clipped}}(t)\}$ are then computed from $\tau_k$'s data only (line 3 to 7). During this step, we perform multiple local updates on the same trajectory's data, leveraging larger global updates without incurring more privacy leakage. This is made possible because the global model is updated with clipped gradients only. We later introduce ensemble-adapted clipping strategies to control the privacy budget over model ensembles, ensuring that the sensitivity of the ensemble gradient $\Delta_k^{\text{clipped}}(t) = \left( \Delta_{i,k}^{\text{clipped}}(t) \right)_{i=1}^{N}$ is bounded by $C$. We then compute an unbiased estimator of the subset gradient average whose sensitivity is bounded by $C/qK$ (line 8). We can then apply the Gaussian mechanism with magnitude $\sigma = zC/qK$, where $z$ controls the strength of the privacy guarantee $\epsilon$, and update the ensemble model $\theta(t) = (\theta_i(t))_{i=1}^{N}$ with noisy gradient (line 9):

$$\theta(t+1) \longleftarrow \theta(t) + \Delta^{\text{avg}}(t) + \mathcal{N} \left( 0_{Nd}, \sigma^2 I_{Nd} \right) \ .$$

---

**Algorithm 1** TDP MODEL ENSEMBLE TRAINING

---

1: **for** each iteration $t \in [\![0, T-1]\!]$ **do**
2:    $\mathcal{U}_t \leftarrow$ (sample with replacement trajectories from $\mathcal{D}_K$ with prob. $q$)
3:    **for** each trajectory $\tau_k \in \mathcal{U}_t$ **do**
4:       Clone current models $\{\theta_i^{\text{start}}\}_{i=1}^N \leftarrow \{\theta_i(t)\}_{i=1}^N$
5:       $\{\theta_{i,k}\}_{i=1}^N \leftarrow \text{ENSCLIPGD}\left(\tau_k, \{\theta_i^{\text{start}}\}_{i=1}^N ; C, \text{local epochs } E, \text{batch size } B\right)$
6:       $\Delta_{i,k}^{\text{clipped}}(t) \leftarrow \theta_{i,k} - \theta_i^{\text{start}}, \;\; i = 1, ..., N$
7:    **end for**
8:    $\Delta_i^{\text{avg}}(t) = \frac{\sum_{k \in \mathcal{U}_t} \Delta_{i,k}^{\text{clipped}}(t)}{qK}, \;\; i = 1, ..., N$
9:    $\theta(t+1) \leftarrow \theta(t) + \Delta^{\text{avg}}(t) + \mathcal{N}\left(0_{Nd}, \left(\frac{zC}{qK}\right)^2 I_{Nd}\right)$
10: **end for**

---

### 4.2.2 PRIVACY GUARANTEES FOR THE MODEL

We can now derive formal privacy guarantees for a model trained using Algorithm 1. A key challenge in our setting arises from training an ensemble of $N$ models for uncertainty estimation, all using the same dataset $\mathcal{D}_K$. Treating each model independently, with separate clipping and noise addition, would be inefficient and significantly increase the privacy budget by composition. This could be mitigated by limiting the ensemble size, but at the cost of performance, as shown in Lu et al. (2022).

To address this challenge, we process all the gradients of the model ensemble simultaneously and distribute the global clipping norm $C$ across all models, on the same principle as the per-layer clipping used in McMahan et al. (2017). Denoting $\Delta_{i,\ell}$ the gradient of layer $\ell$ for model $i$, we propose and experiment with two ensemble clipping strategies: **Flat Ensemble Clipping**, which clips the whole model gradient $\Delta_i = (\Delta_{i,\ell})_{\ell=1}^L$ with $C_i = C/\sqrt{N}$; and **Per Layer Ensemble Clipping**, which clips per-layer gradients $\Delta_{i,\ell}$ with $C_{i,\ell} = C/\sqrt{N \times L}$, so that $C = \sqrt{\sum_{i=1}^N C_i^2} = \sqrt{\sum_{i=1}^N \sum_{\ell=1}^L C_{i,\ell}^2}$.

For both strategies, we verify that that $\Delta_k^{\text{clipped}} = \left(\Delta_{i,k}^{\text{clipped}}\right)_{i=1}^K$ has sensitivity bounded by $C$ (see Theorem 4.2's proof in appendix), and that the contribution of a given trajectory to the *model ensemble* is appropriately limited. Ensemble clipping eliminates the linear dependence of the privacy budget on the number of models. However, it does not entirely remove the negative impact of increasing $N$. Indeed, for a given noise level, a larger $N$ requires a smaller clipping threshold $C_i$ or $C_{i,\ell}$, which can degrade model convergence by losing too much information from the original gradient. Nevertheless, the clipping threshold scales with the square root of $N$, mitigating the impact to some extent.

We now formally derive the privacy guarantees for an ensemble of models trained with Algorithm 1. Mapping users in federating learning to trajectories in offline RL, we can directly adapt Theorem 1 from McMahan et al. (2018) to state that, with the sensitivity of clipped gradients $\Delta_{i,k}^{\text{clipped}}$ effectively bounded by $C$, the moments accounting method from Abadi et al. (2016) computes correctly the privacy loss of Algorithm 1 at trajectory-level for the noise multiplier $z = \sigma/\mathbb{C}$ with $\mathbb{C} = C/qK$. We can therefore use the moments accountant to compute, given $\delta \in (0,1)$, $z > 0$, $q \in (0,1)$ and $T \in \mathbb{N}$, the total privacy budget $\epsilon$ spent by Algorithm 1, and obtain $(\epsilon, \delta)$-TDP guarantees for our dynamics model, as stated in Theorem 4.2 (full proof in appendix).

**Theorem 4.2.** $(\epsilon, \delta)$-TDP guarantees for dynamics model. *Given $\delta \in (0,1)$, noise multiplier $z$, sampling ratio $q$ and number of training iterations $T$, let $\epsilon := \epsilon^{MA}(z, q, T, \delta)$ be the privacy budget computed by the moments accounting method from (Abadi et al. (2016), more details in Section H.6). The dynamics model output by Algorithm 1 is $(\epsilon, \delta)$-TDP.*

### 4.3 POLICY OPTIMIZATION UNDER A PRIVATE MODEL

Now that we learned a private model $\hat{M}$ from offline data, we use it as a simulator of the environment to learn a private policy $\hat{\pi}$ with a model-based policy optimization approach. The use of a private model and the privacy constraints on the end policy introduce additional challenges compared to the non-private case, as demonstrated in Section 4.3.1. We study solutions to mitigate the detrimental

effects of private training on policy performance in Section 4.3.2, before deriving formal privacy guarantees for a policy learned under a private model in Section 4.3.3.

### 4.3.1 IMPACT OF PRIVACY ON POLICY OPTIMIZATION

It is first essential to examine the complexities of policy optimization in model-based offline RL and assess whether they are amplified in the private setting. A major challenge in model-based offline RL is to handle the discrepancy between the true and the learned dynamics when optimizing the policy. Indeed, model inaccuracies cause errors in policy evaluation that may be exploited, resulting in poor performance in the real environment. According to the Simulation Lemma (Kearns & Singh, 2002; Xu et al., 2020), the value evaluation error of a policy $\pi$ in model-based RL can be decomposed into a *model error* term and a *policy distribution shift* term. Formally, denoting $\rho_P^{\pi^B}$ the state-action discounted occupancy measure of the data-collection policy $\pi^B$ under the true MDP, if the model error is bounded as $\mathbb{E}_{(s,a) \sim \rho_P^{\pi^B}} \left[ D_{KL} \left( P(\cdot|s,a) \| \hat{P}(\cdot|s,a) \right) \right] \leq \varepsilon_m$ and the distribution shift is bounded as $\max_s D_{KL} \left( \pi(\cdot|s) \| \pi^B(\cdot|s) \right) \leq \varepsilon_\pi$ , then the value evaluation error of $\pi$ is bounded as:

$$|\hat{V}^\pi - V^\pi| \leq \frac{\sqrt{2}\gamma}{(1-\gamma)^2} \sqrt{\epsilon_m} + \frac{2\sqrt{2}}{(1-\gamma)^2} \sqrt{\varepsilon_\pi} \ , \tag{1}$$

where $\hat{V}^\pi$ and $V^\pi$ denote the value of $\pi$ under the learned and the true dynamics, respectively. Controlling this quantity for an arbitrary $\pi$ is crucial in our setting, as it ensures that the learned MDP is a reasonable simulator of the true environment. Moreover, (1) directly implies a bound on the sub-optimality gap, since $|V^\star - V^{\hat{\pi}}| \leq 2 \sup_\pi |\hat{V}^\pi - V^\pi|$. Under some assumptions regarding the model loss function, Proposition 4.3 states the model error term in terms of the size $N_{\mathcal{D}}$ of the dataset.

**Proposition 4.3.** Value evaluation error in non-private offline MBRL. *Let the model loss function be $L$-Lipschitz and $\Delta$-strongly convex, and assumptions from the simulation lemma hold. There is a stochastic convex optimization algorithm for learning the model and a constant $M$ such that, with probability at least $1 - \alpha$, and for sufficiently large $N_{\mathcal{D}}$, the value evaluation error of $\pi$ is bounded as:*

$$|\hat{V}^\pi - V^\pi| \leq \frac{\sqrt{2}\gamma}{(1-\gamma)^2} \cdot M \cdot \frac{L \log^{1/2}(N_{\mathcal{D}}/\alpha)}{\sqrt{\Delta N_{\mathcal{D}}}} + \frac{2\sqrt{2}\gamma}{(1-\gamma)^2} \sqrt{\varepsilon_\pi} \ .$$

When we learn the model with differential privacy, we disrupt model convergence because of gradient clipping and noise. This likely results in a less accurate dynamics model (although it may help prevent overfitting in some cases) and increased value evaluation error. Intuitively, DP training impacts model error in (1) as a direct result of gradient perturbations: Bassily et al. (2014), in particular, shows that noisy gradient descent (GD) has increased excess risk compared to non-private GD. In the simpler case where the model is trained with a vanilla DP noisy GD algorithm, Proposition 4.4 states the value evaluation error under the private model.

**Proposition 4.4.** Value evaluation error in private offline MBRL. *Let assumptions from Proposition 4.3 hold. If the model is learned with $(\epsilon, \delta)$-DP gradient descent, then, with probability at least $1 - \alpha$, there is a constant $M'$ such that for large enough $N_{\mathcal{D}}$, the value evaluation error of $\pi$:*

$$|\hat{V}_{DP}^\pi - V^\pi| \leq \frac{\sqrt{2}\gamma}{(1-\gamma)^2} \cdot M' \cdot \frac{L d^{1/4} \log(N_{\mathcal{D}}/\delta) \cdot poly \log(1/\alpha)}{\sqrt{\Delta N_{\mathcal{D}} \epsilon \alpha}} + \frac{2\sqrt{2}\gamma}{(1-\gamma)^2} \sqrt{\varepsilon_\pi} \ ,$$

where $\hat{V}_{\text{DP}}^\pi$ is the value of $\pi$ under the privately learned dynamics. Comparing the value evaluation errors in Propositions 4.3 and 4.4 (both proven in appendix), we observe how DP training may degrade performance in MBRL. The private bound has an explicit dependence on the problem dimension $d$ which is not present in the non-private bound, and the $\sqrt{\epsilon}$ factor in the denominator shows that the error will degrade with strong privacy guarantees. On the other hand, the distribution shift term does not depend on the learned dynamics and is therefore not affected by private training.

### 4.3.2 MITIGATING PRIVATE MODEL UNCERTAINTY

In Section 4.3.1, we showed that private model training impacts the reliability of our model for evaluating policies due to an increased dynamics error, which can lead to misjudging the quality of a

policy in the true environment. In the non-private case, this is typically handled by penalizing the reward with a measure of the uncertainty of the model, denoted $u : \mathcal{S} \times \mathcal{A} \to \mathbb{R}_+$. Therefore, if the model is believed to be unreliable at a given state-action pair $(s, a)$ (*i.e.*, large $u(s, a)$), the possibly over-estimated reward will be corrected as:

$$\tilde{r}(s, a) = \hat{r}(s, a) - \lambda \cdot u(s, a) \quad , \tag{2}$$

where $\lambda$ is an hyperparameter. The policy is then optimized under the resulting pessimistic MDP $\tilde{\mathcal{M}} = (\mathcal{S}, \mathcal{A}, \hat{P}, \tilde{r}, \gamma, \rho_0)$. MOPO (Yu et al., 2020), MOREL (Kidambi et al., 2020) and more recently COUNT-MORL (Kim & Oh, 2023) achieve impressive results on traditional offline RL benchmarks with this approach, using different heuristics to estimate model uncertainty.

As suggested by the simulation lemma, the valuation error can depend on both model error and distribution shift. However, we demonstrated that private training affects only model error. Interestingly, Lu et al. (2022), which studies design choices in offline model-based RL and the properties of various uncertainty estimators, finds that the uncertainty measures proposed in the literature are more strongly correlated with model error than with distribution shift. Based on this, we believe that existing uncertainty measures are well-suited to mitigate the diminished reliability of the model under private training, as they will effectively capture the increased error. In particular, we consider the maximum aleatoric uncertainty $u_{\text{MA}}(s, a) = \max_{i \in [\![1,N]\!]} \|\Sigma_{\psi_i}(s, a)\|_F$ (Yu et al., 2020) and the maximum pairwise difference $u_{\text{MPD}}(s, a) = \max_{i,j \in [\![1,N]\!]} \|\mu_{\phi_i}(s, a) - \mu_{\phi_j}(s, a)\|_2$ (Kidambi et al., 2020). We compare both estimators (see Table 10 in the appendix) and find that neither is consistently superior. However, we observe that the choice of estimator can affect performance on a specific task. In addition, it seems reasonable to moderately increase the reward penalty $\lambda$ compared to the non-private case to take into account the greater uncertainty.

### 4.3.3 PRIVATE POLICY OPTIMIZATION

Given a choice of uncertainty estimator $u \in \{u_{\text{MA}}, u_{\text{MPD}}\}$, we now consider optimizing the policy within the pessimistic private MDP $\tilde{\mathcal{M}} = (\mathcal{S}, \mathcal{A}, \hat{P}, \tilde{r}_u, \gamma, \rho_0)$, with $\tilde{r}_u = \hat{r}(s, a) - \lambda \cdot u(s, a)$. We use Soft Actor-Critic (SAC, Haarnoja et al. (2018))[1], a classic off-policy algorithm with entropy regularization, to learn the policy from $\tilde{\mathcal{M}}$, in line with existing approaches in the offline MBRL literature. Offline model-based methods typically mix real offline data from $\mathcal{D}_K$ with model data during policy learning (in MOPO, for instance, each batch contains 5% of real data). Here, however, we learn the policy exclusively from model data to avoid incurring privacy loss beyond what is needed to train the model, and thus control the privacy guarantees. Algorithm 4 in appendix provides a pseudo-code for SAC policy optimization in the pessimistic private MDP. Using the post-processing property of DP, we can now state in Theorem 4.5 that, given the $(\epsilon, \delta)$-TDP model $\hat{M} = (\hat{P}, \hat{r})$ learned as described in Section 4.2, the policy learned with Algorithm 4 under $\tilde{M}$ is also $(\epsilon, \delta)$-TDP. The full proof of this theorem is provided in appendix.

**Theorem 4.5.** $(\epsilon, \delta)$-*TDP guarantees for* PRIMORL. *Given an* $(\epsilon, \delta)$-*TDP model* $\left(\hat{P}, \hat{r}\right)$ *learned with Algorithm 1, the policy obtained with private policy optimization (Algorithm 4) within the pessimistic model* $\left(\hat{P}, \hat{r} - \lambda u\right)$ *is* $(\epsilon, \delta)$-*TDP.*

## 5 EXPERIMENTS

We empirically assess PRIMORL in three continuous control tasks: CARTPOLE-BALANCE and CARTPOLE-SWINGUP from the DeepMind Control Suite (Tassa et al., 2018) as well as PENDULUM from OpenAI's Gym (Brockman et al., 2016). We also conduct experiments on HALFCHEETAH (Wawrzynski, 2009), which we present in appendix (Section J). For simplicity, we refer to CARTPOLE-BALANCE and CARTPOLE-SWINGUP as BALANCE and SWINGUP.

### 5.1 EXPERIMENTAL SETTING

Following common practice, we evaluate the offline policies by running them in the real environment We aim to assess the policy's performance degradation when varying the privacy level, as DP training

---

[1]This could be any model-based policy optimization or planning algorithm that does not use offline data.

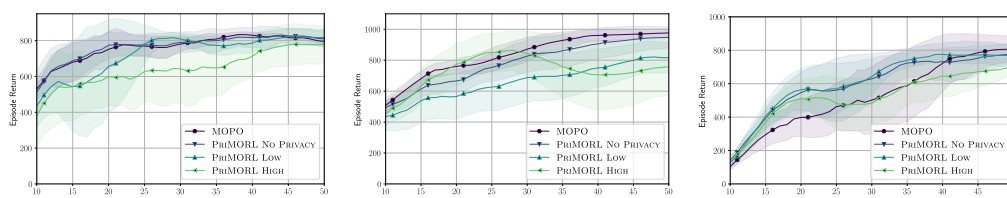

Figure 2: Learning curves on PENDULUM (*left*), BALANCE (*middle*) and SWINGUP (*right*).

may negatively affect it. We consider MOPO as our non-private baseline. For PRIMORL, we consider different configurations outlined in Table 3. The NO PRIVACY variant, without noise ($z = 0$), isolates the impact of trajectory-level model training on performance. The two private variants ($\epsilon < \infty$) PRIMORL LOW and PRIMORL HIGH correspond to different noise multipliers. We discuss the choice of privacy parameters, as well as training hyperparameters and implementation details in appendix (Section H).

As the existing SWINGUP offline benchmark from Gülçehre et al. (2020) is very small ($K = 40$), and DP training of ML models typically requires significantly more data compared to non-private training (see, for instance, Ponomareva et al. (2023), and our discussion in appendix, Section L), we build our own dataset with 30k trajectories (*i.e.*, 30M steps). We follow the same approach for BALANCE and PENDULUM for which we are not aware of any existing offline benchmark. Data collection, detailed in appendix (Section D), follows the philosophy of standard benchmarks like D4RL (Fu et al., 2020).

## 5.2 MAIN RESULTS

We present results on BALANCE, SWINGUP and PENDULUM for PRIMORL and baselines in Table 1 and Figure 2. Both report policy performance in the real MDP as the mean episodic return over 10 episodes per SAC training epoch. Average performance and 95% confidence intervals are computed by re-training the model and the policy from scratch on at least 5 random seeds to assess the stability of the full training process. We also report the corresponding theoretical upper bound on $\epsilon$, as computed from the hyperparameters $z, q, T$ and $\delta$ using the moments accountant $\epsilon^{\text{MA}}(z, q, T, \delta)$ (see Table 2 and discussion in Section H.6 for further explanations regarding the moments accountant).

These results show a well-expected trade-off: performance tends to degrade with stronger privacy guarantees (*i.e.*, smaller $\epsilon$'s), as the model training gets perturbed with higher levels of noise. Moreover, private model training makes the policy performance less stable over several runs, which is also expected since differential privacy adds another source of randomness during training. We notice that noise is not the sole factor that negatively impacts performance, as suggested by the gap between MOPO and PRIMORL NO PRIVACY: gradient clipping and trajectory-level training also contribute to performance degradation. In some cases, a small amount of DP noise might actually be beneficial, acting as a kind of regularization, as in SWINGUP and PENDULUM. Moreover, experiments on HALFCHEETAH (Section J) show that PRIMORL performs worse in higher-dimensional tasks. This could be expected based on the theoretical analysis led in Section 4.3.1, as DP training adds a dependence on the dimension $d$ of the task in the valuation gap. Despite this trade-off, private agents trained with PRIMORL remain competitive with MOPO for $\epsilon$ in the $10^1$ to $10^2$ range. For PENDULUM, we plot policy performance against $\epsilon$ in Figure 3, and observe even no performance degradation until $\epsilon$ reaches the 1 to 10 range. Although algorithms from Qiao & Wang (2023a) are not suited for direct comparison on the same tasks, we argue that our empirical results are significantly stronger. Indeed, converting $\rho$-zero-concentrated DP guarantees into standard $(\epsilon, \delta)$-DP guarantees for clarity and fair comparison, we observe that PRIMORL achieves comparable privacy-performance trade-offs, but on much more complex environments (more details in appendix, Section F).

While the privacy budgets $\epsilon$ from Table 1 do not correspond to strong theoretical privacy guarantees, we must consider the worst-case nature of the differential privacy definition, along with its very strong assumptions on the adversary side. In offline RL especially, the definition of DP assumes the adversary only has to discriminate between two precise neighboring datasets $D$ and $D' = D \cup \{\tau\}$ as well as the release of all gradients, whereas in practice the adversary faces the much harder task of reconstructing a high-dimensional trajectory based on the output policy and limited side information

Table 1: Results for PENDULUM, BALANCE and SWINGUP.

| METHOD | PENDULUM | | CARTPOLE-BALANCE | | CARTPOLE-SWINGUP | |
|---|---|---|---|---|---|---|
| | $\epsilon$ | RETURN | $\epsilon$ | RETURN | $\epsilon$ | RETURN |
| MOPO | $\infty$ | $795.9 \pm 6.5$ | $\infty$ | $976.3 \pm 26.8$ | $\infty$ | $804.9 \pm 89.6$ |
| PRIMORL NO PRIV. | $\infty$ | $810.4 \pm 27.5$ (**101.8%**) | $\infty$ | $947.5 \pm 68.3$ (**97.1%**) | $\infty$ | $774.1 \pm 81.7$ (**96.17%**) |
| PRIMORL LOW | 22.3 | $817.4 \pm 21.7$ (**102.7%**) | 85.0 | $815.8 \pm 97.2$ (**83.6%**) | 94.2 | $772.4 \pm 73.9$ (**95.96%**) |
| PRIMORL HIGH | 5.1 | $778.9 \pm 53.5$ (**97.9%**) | 8.2 | $758.2 \pm 187.2$ (**77.7%**) | 17.0 | $698.3 \pm 57.5$ (**86.75%**) |

| | | $z$ | $T$ | $q$ | $\delta$ | $\epsilon$ |
|---|---|---|---|---|---|---|
| PENDULUM | LOW | 0.35 | $7.10^3$ | $10^{-3}$ | $10^{-5}$ | 22.3 |
| | HIGH | 0.52 | $7.10^3$ | $10^{-3}$ | $10^{-5}$ | **5.1** |
| BALANCE | LOW | 0.25 | $7.10^3$ | $10^{-3}$ | $10^{-5}$ | 85.0 |
| | HIGH | 0.45 | $7.10^3$ | $10^{-3}$ | $10^{-5}$ | **8.2** |
| SWINGUP | LOW | 0.25 | $10.10^3$ | $10^{-3}$ | $10^{-5}$ | 94.2 |
| | HIGH | 0.38 | $10.10^3$ | $10^{-3}$ | $10^{-5}$ | **17.0** |

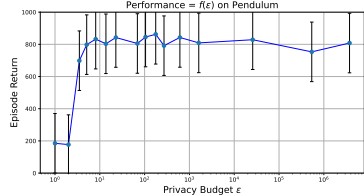

Table 2: Hyperparameters and computation of the theoretical privacy budget $\epsilon := \epsilon^{\mathrm{MA}}(z, q, T, \delta)$

Figure 3: Policy performance on PENDULUM as a function of $\epsilon$

only. Therefore, backed by recent work on empirical privacy auditing (*e.g.*, Carlini et al. (2019); Ponomareva et al. (2022)), we argue that such $\epsilon$'s can provide adequate privacy protection in practical offline RL applications. According to Ponomareva et al. (2023), $\epsilon \lesssim 10$ is actually a realistic and widely used goal in private deep learning applications. We discuss this matter more in depth in appendix (Section G). We also point out that achieving a strong privacy-utility trade-off in offline RL requires access to datasets with a very large number of trajectories and that current benchmarks, with datasets of only dozens to thousands of trajectories, are insufficient for studying privacy effectively. In contrast, other fields often use datasets containing millions of users (between $10^6$ to $10^9$ users in McMahan et al. (2018)) to ensure robust privacy guarantees, which would be very costly to study in offline RL. In appendix (Section L), we provide evidence that increasing dataset size improves the privacy-performance trade-off, demonstrating even greater potential for PRIMORL.

# 6 DISCUSSION

While existing DP RL methods are limited to tabular and linear finite-horizon MDPs, we are the first to address deep offline RL with privacy guarantees in the infinite-horizon discounted setting, and propose a model-based approach named PRIMORL. We empirically show that PRIMORL is capable of learning trajectory-level private, neural-based policies in standard control tasks with only limited performance cost, achieving a new standard in differentially private RL. Although the reported privacy budgets are typically considered too large to stand as formal DP guarantees, we argue based on recent studies on practical DP that they can offer satisfying privacy protection in practice, especially considering the worst-case nature of DP which can yield too pessimistic privacy budgets. Empirical evaluation of the robustness of our algorithm against privacy attacks, for which a rigorous and standardized benchmark has to be developed, will thus be an important research direction for future work. We further point out that our approach has the potential for achieving greater privacy-utility trade-offs given access to large enough offline datasets, hence calling for new benchmarks in the increasingly important field of private offline RL.

With the aim of shifting the paradigm in how private RL is approached — from predominantly theoretical research to practical algorithms — this work sets the stage for future efforts to scale to higher-dimensional problems. We identify several promising research avenues. First, we may consider limiting the number of real trajectories used during training to leverage privacy amplification through sub-sampling, for example, by using data augmentation techniques. As our theoretical analysis highlights the impact of dimensionality on private model error, another promising direction is learning compact representations of high-dimensional inputs and performing planning directly in the latent space, as explored by Jiang et al. (2023). We leave these avenues for future work. Overall, we believe our work represents a significant step toward the much-needed deployment of private RL methods in practical applications.

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

# A PROOFS

**Theorem 4.2.** $(\epsilon, \delta)$-*TDP guarantees for dynamics model. Given $\delta \in (0, 1)$, noise multiplier $z$, sampling ratio $q$ and number of training iterations $T$, let $\epsilon := \epsilon^{MA}(z, q, T, \delta)$ be the privacy budget computed by the moments accounting method from (Abadi et al. (2016), more details in Section H.6). The dynamics model output by Algorithm 1 is $(\epsilon, \delta)$-TDP.*

*Proof.* Theorem 1 from McMahan et al. (2018) shows that the moments accounting method from Abadi et al. (2016) computes correctly the privacy loss of DP-FEDAVG at user-level for the noise multiplier $z = \sigma/\mathbb{C}$ with $\mathbb{C} = C/qK$ if, for each user $u_k$, the clipped gradient $\Delta_k^{\text{clipped}}$ computed from $u_k$'s data has sensitivity bounded by $C$ (referred to as **Condition 1**). With TDP MODEL ENSEMBLE TRAINING, we train the model ensemble as a single big model: at each training iteration, the same input batch is processed forward by all models in a single pass, a single loss is computed for the ensemble, and the parameters are then updated in a single backward pass. The ensemble of models can therefore be seen as a concatenation of all individual models, equivalent to a larger model $\theta = (\theta_i)_{i=1}^N$. We can therefore extend this theorem by mapping users in federating learning to trajectories in offline RL, as long as **Condition 1** holds for every trajectory $\tau_k$.

Since we use **ensemble clipping**, we verify that, for trajectory $\tau_k$, the ensemble gradient $\Delta_k^{\text{CLIPPED}} = \left(\Delta_{i,k}^{\text{CLIPPED}}\right)$ has sensitivity bounded by $C$. With **flat ensemble clipping**, the gradient of each model $i \in [\![1, N]\!]$ is clipped by a factor $C_i = \frac{C}{\sqrt{N}}$ (see Algorithm 3). By construction, $\Delta_{i,k}^{\text{CLIPPED}}$ has sensitivity bounded by $C_i$, *i.e.*, $\max_{d(D,D')=1} \|\Delta_{i,k}^{\text{CLIPPED}}(D) - \Delta_{i,k}^{\text{CLIPPED}}(D')\|_2 \leq C_i$. Therefore, for two neighboring datasets $D$ and $D'$:

$$\|\Delta_k^{\text{CLIPPED}}(D) - \Delta_k^{\text{CLIPPED}}(D')\|_2 = \| \left(\Delta_k^{\text{CLIPPED}}(D) - \Delta_k^{\text{CLIPPED}}(D')\right)_{i=1}^N \|_2$$

$$= \sqrt{\sum_{i=1}^N \|\Delta_{i,k}^{\text{CLIPPED}}(D) - \Delta_{i,k}^{\text{CLIPPED}}(D')\|_2^2}$$

$$\leq \sqrt{\sum_{i=1}^N C_i^2}$$

$$= \sqrt{\sum_{i=1}^N \frac{C^2}{N}}$$

$$= C \ .$$

This implies $\max_{d(D,D')=1} \|\Delta_k^{\text{CLIPPED}}(D) - \Delta_k^{\text{CLIPPED}}(D')\|_2 \leq C$: $\Delta_k^{\text{CLIPPED}}$ has sensitivity bounded by $C$. We can derive the same proof for **per-layer ensemble clipping**. Therefore, Theorem 1 from McMahan et al. (2018) holds for TDP MODEL ENSEMBLE TRAINING.

We can therefore use the moments accountant $\epsilon^{\text{MA}}$ to compute, given $z > 0$, $\delta \in (0, 1)$, $q \in (0, 1)$ and $T \in \mathbb{N}$, the total privacy budget $\epsilon$ spent by Algorithm 1, *i.e.*, $\epsilon = \epsilon^{\text{MA}}(z, q, T, \delta)$.

The dyanmics model output by Algorithm 1 is therefore $(\epsilon, \delta)$-TDP. $\qquad\square$

**Theorem 4.5.** $(\epsilon, \delta)$-*TDP guarantees for PRIMORL. Given an $(\epsilon, \delta)$-TDP model $\left(\hat{P}, \hat{r}\right)$ learned with Algorithm 1, the policy obtained with private policy optimization (Algorithm 4) within the pessimistic model $\left(\hat{P}, \hat{r} - \lambda u\right)$ is $(\epsilon, \delta)$-TDP.*

*Proof.* First, we establish that the pessimistic MDP $\tilde{M}$ is private for $u \in \{u_{\text{MA}}, u_{\text{MPD}}\}$. By Theorem 4.2, both the mean estimators $\{\mu_{\phi_i}\}_{i=1}^N$ the covariance estimators $\{\Sigma_{\psi_i}\}_{i=1}^N$ are private. Therefore, both uncertainty estimators $u_{\text{MA}}(s, a) = \|\Sigma_{\psi_i}(s, a)\|_F$ and $u_{\text{MPD}}(s, a) = \max_{i,j} \|f_{\phi_i} - f_{\phi_j}\|_2$, as data-independent transformations of the above quantities, are also private thanks to the post-processing property of DP. Therefore, the pessimistic model $\tilde{M}$ remains $(\epsilon, \delta)$-TDP.

Now, we can think of SAC model-based policy optimization (Algorithm 4) as an abstract, randomized function $h_\Pi$, that takes as input $\hat{M}$ and outputs as policy $\hat{\pi}$. Furthermore, let $h_M$ denote the mechanism that takes as input the private offline dataset $\mathcal{D}_K$ and outputs the private pessimistic model $\hat{M}$, and which is $(\epsilon, \delta)$-TDP following 4.2. We observe that $h = h_\Pi \circ h_M$, where $h$ is the global offline RL algorithm which is the object of Definition 4.1. Since SAC only uses data from the model, as stated in Section 4.3.3, $h_\Pi$ is independent of the private offline data $\mathcal{D}_K$. In other words, $h_\Pi$ is a data-independent transformation of the private mechanism $h_M$. Thanks again to the post-processing property of differential privacy, $h$ is also $(\epsilon, \delta)$-TDP. □

We now prove the following two propositions:

**Proposition 4.3.** Value evaluation error in non-private offline MBRL. *Let the model loss function be $L$-Lipschitz and $\Delta$-strongly convex, and assumptions from the simulation lemma hold. There is a stochastic convex optimization algorithm for learning the model and a constant $M$ such that, with probability at least $1 - \alpha$, and for sufficiently large $N_\mathcal{D}$, the value evaluation error of $\pi$ is bounded as:*

$$|\hat{V}^\pi - V^\pi| \leq \frac{\sqrt{2}\gamma}{(1-\gamma)^2} \cdot M \cdot \frac{L \log^{1/2}(N_\mathcal{D}/\alpha)}{\sqrt{\Delta N_\mathcal{D}}} + \frac{2\sqrt{2}\gamma}{(1-\gamma)^2}\sqrt{\varepsilon_\pi} \ .$$

**Proposition 4.4.** Value evaluation error in private offline MBRL. *Let assumptions from Proposition 4.3 hold. If the model is learned with $(\epsilon, \delta)$-DP gradient descent, then, with probability at least $1 - \alpha$, there is a constant $M'$ such that for large enough $N_\mathcal{D}$, the value evaluation error of $\pi$:*

$$|\hat{V}^\pi_{DP} - V^\pi| \leq \frac{\sqrt{2}\gamma}{(1-\gamma)^2} \cdot M' \cdot \frac{Ld^{1/4} \log(N_\mathcal{D}/\delta) \cdot poly \log(1/\alpha)}{\sqrt{\Delta N_\mathcal{D}\epsilon\alpha}} + \frac{2\sqrt{2}\gamma}{(1-\gamma)^2}\sqrt{\varepsilon_\pi} \ ,$$

*Proof.* Let $\mathcal{F}$ denote the function class of the model. The model is estimated by maximizing the likelihood of the data $\mathcal{D}_K = (s_i, a_i, s'_i)_{i=1}^N$, which is collected by an unknown behavioral policy $\pi^B$. This is equivalent to minimizing the negative log-likelihood. The population risk of the estimated model $\hat{P}$ obtained with DP-SGD, is therefore:

$$\mathcal{L}(\hat{P}) = \mathbb{E}_{(s,a)\sim\rho_P^{\pi^B}, s'\sim P(\cdot|s,a)} \left[ -\log \hat{P}(s'|s,a) \right] \ ,$$

where $\rho_P^{\pi^B}$ is the (normalized) state-action occupancy measure under policy $\pi^B$ and dynamics $P$.

Let us further assume that the true model $P$ belongs to the function class $\mathcal{F}$, and that $P \in \operatorname{argmin}_{P'\in\mathcal{F}}\mathcal{L}(P')$. We can therefore write the excess population risk of the model estimator $\hat{P}$ as:

$$\mathcal{L}(\hat{P}) - \mathcal{L}(P) = \mathbb{E}_{(s,a)\sim\rho_P^{\pi^B}, s'\sim P(\cdot|s,a)} \left[ \frac{\log P(s'|s,a)}{\log \hat{P}(s'|s,a)} \right] \ .$$

But, denoting $D_{\mathrm{KL}}(A, B)$ the Kullback-Leibler divergence between distributions $A, B$:

$$D_{\mathrm{KL}}\left( P(s,a), \hat{P}(s,a) \right) = \mathbb{E}_{s'\sim P(\cdot|s,a)} \left[ \frac{\log P(s'|s,a)}{\log \hat{P}(s'|s,a)} \right] \ .$$

We can therefore rewrite the above excess population risk as:

$$\mathcal{L}(\hat{P}) - \mathcal{L}(P) = \mathbb{E}_{(s,a)\sim\rho_P^{\pi^B}} \left[ D_{\mathrm{KL}}\left( P(s,a), \hat{P}(s,a) \right) \right] \ . \tag{3}$$

If the objective function $\mathcal{L}$ is $L$-Lipschitz and $\Delta$-strongly convex, Bassily et al. (2014) shows (Theorem F.2) that, given $N_\mathcal{D}$ data points, a noisy gradient descent algorithm with $(\epsilon, \delta)$-DP guarantees satisfies, with probability at least $1 - \alpha$:

$$\mathcal{L}(\hat{P}) - \mathcal{L}(P) = \mathcal{O}\left( \frac{L^2\sqrt{d}\log^2(N_\mathcal{D}/\delta) \cdot \operatorname{poly}\log(1/\alpha)}{\Delta N_\mathcal{D}\epsilon\alpha} \right) \ . \tag{4}$$

In the non-private case, Shalev-Shwartz et al. (2009) provides the following bound under the same assumptions:

$$\mathcal{L}(\hat{P}) - \mathcal{L}(P) = \mathcal{O}\left( \frac{L^2\log(N_\mathcal{D}/\alpha)}{\Delta N_\mathcal{D}} \right) \ . \tag{5}$$

On the other hand, we have from the Simulation Lemma (Kearns & Singh, 2002; Xu et al., 2020) that for a MDP $\mathcal{M}$ with reward upper bounded by $r_{\max} = 1$ and dynamics $P$, a behavioral policy $\pi^B$ and a learned transition model $\hat{P}$ with:

$$\mathbb{E}_{(s,a) \sim \rho_P^{\pi^B}} \left[ D_{\text{KL}} \left( P(s,a), \hat{P}(s,a) \right) \right] \leq \varepsilon_M \quad , \tag{6}$$

which by 3 is equivalent to:

$$\mathcal{L}(\hat{P}) - \mathcal{L}(P) \leq \varepsilon_M \quad , \tag{7}$$

Let $\pi$ be an arbitrary policy. If the divergence between $\pi$ and the behavioral policy is bounded:

$$\max_s D_{\text{KL}} \left( \pi(\cdot|s), \pi^B(\cdot|s) \right) \leq \varepsilon_\pi \quad , \tag{8}$$

then the value evaluation error of $\pi$ is bounded as:

$$|\hat{V}^\pi - V^\pi| \leq \frac{\sqrt{2}\gamma}{(1-\gamma)^2} \sqrt{\varepsilon_M} + \frac{2\sqrt{2}\gamma}{(1-\gamma)^2} \sqrt{\varepsilon_\pi} \quad . \tag{9}$$

Since $f(x) = \mathcal{O}(g(x))$ implies $\sqrt{f(x)} = \mathcal{O}(\sqrt{g(x)})$ [2], we note that we can replace $\sqrt{\varepsilon_M}$ in the model term of the right-hand side of 9 by the (square root of) the bounds from 4 and 5 in the private case and in the non-private case, respectively.

This result holds for any policy $\pi$ verifying 8. In particular, if:

$$\max_s D_{\text{KL}} \left( \hat{\pi}(\cdot|s), \pi^B(\cdot|s) \right) \leq \varepsilon_{\hat{\pi}} \quad , \tag{10}$$

then:

$$|\hat{V}^{\hat{\pi}} - V^{\hat{\pi}}| \leq \frac{\sqrt{2}\gamma}{(1-\gamma)^2} \sqrt{\varepsilon_M} + \frac{2\sqrt{2}\gamma}{(1-\gamma)^2} \sqrt{\varepsilon_{\hat{\pi}}} \quad . \tag{11}$$

$\square$

---

[2]Indeed, for $f(x)$ positive, for any $x \geq x_0$, $|f(x)| = f(x) \leq M' \times g(x)$, then, for any $x \geq x_0$, $\sqrt{f(x)} = |\sqrt{f(x)}| \leq \sqrt{M'} \times \sqrt{g(x)} = M \times \sqrt{g(x)}$

# B RELATED WORK (EXTENDED)

## B.1 MODEL-BASED OFFLINE REINFORCEMENT LEARNING

Unlike classical RL (Sutton & Barto, 1998) which is online in nature, offline RL (Levine et al., 2020; Prudencio et al., 2022) aims at learning and controlling autonomous agents without further interactions with the system. This approach is preferred or even unavoidable in situations where data collection is impractical (see for instance Singh et al. (2022); Liu et al. (2020); Kiran et al. (2022)). Model-based RL (Moerland et al., 2023) can also help when data collection is expensive or unsafe as a good model of the environment can generalize beyond in-distribution trajectories and allow simulations. Moreover, model-based RL has been shown to be generally more sample efficient than model-free RL (Chua et al., 2018). Argenson & Dulac-Arnold (2021) also show that model-based offline planning, where the model is learned offline on a static dataset and subsequently used for control without further accessing the system, is a viable approach to control agents on robotic-like tasks with good performance. Unfortunately, the offline setting comes with its own major challenges. In particular, when the data is entirely collected beforehand, we are confronted to the problem of *distribution shift* (Fujimoto et al., 2019): as the logging policy used to collect the training dataset only covers a limited (and potentially small) region of the state-action space, the model can only be trusted in this region, and may be highly inaccurate in other parts of the space. This can lead to a severe decrease in the performance of classic RL methods, particularly in the model-based setting where the acting agent may exploit these inaccuracies in the model , causing large gap between performances in the true and the learned environment. MOPO (Yu et al., 2020) and MOREL (Kidambi et al., 2020), and more recently COUNT-MORL (Kim & Oh, 2023) have effectively tackled this issue by penalizing the reward proportionally to the model's uncertainty, achieving impressive results on popular offline benchmarks. Nonetheless, there remain many areas for improvement, as highlighted by Lu et al. (2022), which extensively study and challenge key design choices in offline MBRL algorithms.

## B.2 PRIVACY IN REINFORCEMENT LEARNING

Differential Privacy (DP), first formalized in Dwork (2006), has become the gold standard in terms of privacy protection. Over the recent years, the design of algorithms with better privacy-utility trade-offs has been a major line of research. In particular, relaxations of differential privacy and more advanced composition tools have allowed tighter analysis of privacy bounds (Dwork et al., 2010; Dwork & Rothblum, 2016; Bun & Steinke, 2016; Mironov, 2017a). Leveraging these advances, the introduction of DP-SGD (Abadi et al., 2016) has allowed to design private deep learning algorithms, paving the way towards a wider adoption of DP in real-world settings, although the practicalities of differential privacy remain challenging (Ponomareva et al., 2023). In parallel to the theoretical analysis of privacy, many works have focused on designing more and more sophisticated attacks, justifying further the need to design DP algorithms ((Rigaki & Garcia, 2020)).

Recent works on RL-specific attacks (Pan et al., 2019; Prakash et al., 2022; Gomrokchi et al., 2023) have demonstrated that reinforcement learning (RL) is no more immune to privacy threats. With RL being increasingly used to provide personalized services (den Hengst et al., 2020), which may expose sensitive user data, developing privacy-preserving techniques for training policies has become crucial. Shortly after DP was successfully extended to multi-armed bandits (Tossou & Dimitrakakis, 2016; Basu et al., 2019), a substantial body of work (*e.g.*, Vietri et al. (2020); Garcelon et al. (2021); Liao et al. (2021); Luyo et al. (2021); Chowdhury & Zhou (2021); Zhou (2022); Ngo et al. (2022); Qiao & Wang (2023b)) addressed privacy in online RL, extending definitions from bandits. However, relying on count-based and UCB-like methods, current RL algorithms with formal DP guarantees are essentially limited to episodic tabular or linear MDPs, and have not been assessed empirically beyond simple numerical simulations. However, current RL algorithms with formal DP guarantees are essentially limited to episodic tabular or linear MDPs, and have not been assessed empirically beyond simple numerical simulations. Few works have proposed private RL methods for more general problems, however with significant limitations or in different contexts. Wang & Hegde (2019) tackle continuous state spaces by adding functional noise to Q-Learning, but the approach is restricted to unidimensional states and focuses on protecting reward information. Recently, Cundy et al. (2024) addressed high-dimensional control and robotic tasks; however, they consider a specific notion of privacy that protects sensitive state variables based on a mutual information framework.

Despite the relevance of the setting for real-world RL deployments, private offline RL has received comparatively less attention. To date, only Qiao & Wang (2023a) have proposed DP offline algorithms, building on non-private value iteration methods. While their approach lays the groundwork for private offline RL and offers strong theoretical guarantees, it remains limited to episodic tabular and linear MDPs. Consequently, no existing work has introduced DP methods that can handle deep RL environments in the infinite-horizon discounted setting, a critical step toward deploying private RL algorithms in real-world applications. With this work, we aim to fill this gap by proposing a differentially private, deep model-based RL method for the offline setting.

## C    PRESENTATION OF THE TASKS

CARTPOLE requires to swing up then balance an unactuated pole by applying forces on a cart at its base, while CARTPOLE-BALANCE only requires keeping balance. The duration of both tasks is 1,000 steps. PENDULUM involves controlling an inverted pendulum by applying torque to keep it upright and balanced over 200 steps. For this task, we normalize the episodic return to obtain a normalized score between 0 and 1000, using the following formula: $s_{\text{normalized}} = \frac{s-(-1500)}{0-(-1500))}$. HALFCHEETAH is another, higher-dimensional, continuous control task from OpenAI's Gym (Brockman et al., 2016) based on the physics engine MuJoCo (Todorov et al., 2012) where we move forward a 2D cat-like robot by applying torques on its joints. The duration of an episode is 1,000 steps.

## D    DATA COLLECTION

To collect our offline dataset for CARTPOLE and PENDULUM, we used DDPG (Lillicrap et al., 2016), a model-free RL algorithm for continuous action spaces. We ran 600 independent runs of $50,000$ steps each for CARTPOLE-BALANCE, 150 independent runs of $200,000$ steps each for CARTPOLE-SWINGUP, and 6 independent runs of $1M$ steps each for PENDULUM. We collect all training episodes to ensure a correct mix between random, medium and expert episodes (similar to *replay* datasets in Fu et al. (2020)).

## E    BASELINES

Table 3: PRIMORL configurations.

| VARIANT | TRAJECTORY-LEVEL ENS. TRAINING | CLIP | NOISE | DP |
|---------|:---:|:---:|:---:|:---:|
| NO CLIP | ✓ | ✗ | ✗ | $\epsilon = \infty$ |
| NO PRIVACY | ✓ | ✓ | ✗ | $\epsilon = \infty$ |
| LOW, HIGH | ✓ | ✓ | ✓ | $\epsilon < \infty$ |

The first two baselines, PRIMORL NO CLIP and PRIMORL NO PRIVACY are not private ($\epsilon < \infty$) but allow us to isolate the impact of trajectory-level model ensemble training (without clipping and noise addition) and clipping on policy performance. We do not report results for PRIMORL NO CLIP for CARTPOLE and PENDULUM as we found that the model optimized with TDP MODEL ENSEMBLE TRAINING diverges without clipping.

## F    COMPARISON TO EXISTING METHODS

The closest and only comparable work in offline DPRL is Qiao & Wang (2023a). In Table 4, we compare the characteristics of PRIMORL with their algorithm DP-VAPVI (since their other algorithm is only for tabular MDPs and obviously does not compare). This comparison highlights that PRIMORL and DP-VAPVI are designed for very distinct settings. We cannot efficiently implement DP-VAPVI on our benchmark, in particular because of the continuous action spaces and the fact that it explicitly relies on a finite, relatively small horizon $H$ (not only the number of statistics to maintain and privatize depends on $H$ but the amount of noise needed to privatize each statistic also grows linearly with $H$).

Although their scope is limited to finite, tabular and linear MDPs and their algorithms are not suited for direct comparison on the same benchmarks, we provide below a side-by-side comparison of our respective empirical results, with the aim of re-contextualizing our results within the current state of the literature.

First, we compare the complexity of the benchmark tasks considered here and the evaluation environment used in Qiao & Wang (2023a). Qiao & Wang (2023a) evaluate their algorithms on an episodic synthetic linear MDP with 2 states and 100 actions, and horizon $H = 20$. On the other hand, we consider standard control tasks with multi-dimensional continuous state and action spaces. Moreover,

Table 4: Comparison between PRIMORL (Ours) and DP-VAPVI (Qiao & Wang)

|  | PRIMORL (Ours) | DP-VAPVI (Qiao & Wang) |
|---|---|---|
| MODEL-BASED | ✓ | ✓ |
| SETTING | $\gamma$-discounted infinite horizon | Finite horizon $H$ |
| FUNCTION APPROXIMATION | General function approximation, including NN | Linear function approximation with known features $\phi(s, a)$ |
| SPACES | Continuous $\mathcal{S}$ and $\mathcal{A}$ | Could theoretically handle continuous actions, but $\arg\max_{\pi_h(\cdot\|s)} \langle \hat{Q}_h(s, a), \pi_h(\cdot\|s) \rangle$ is impractical to compute for large or infinite $\mathcal{A}$ |
| MODEL TYPE | Global model ensemble (step independent), set of weights $\theta = \{\theta_i\}_i = 1^N$ with $\theta_i \in \mathbb{R}^d$ | Step-dependent model represented by $5H$ statistics |
| PRIVACY BUDGET | Scales with training hyperparameters $q$, $T$, $N$ (indirectly) | Scales with horizon $H$, a parameter of the problem |

our tasks have long horizons and high frequency, which makes them impractical to represent in the episodic setting, justifying the use of the $\gamma$-discounted infinite-horizon setting.

We then compare the privacy-performance trade-offs achieved by Qiao & Wang (2023a) and PRI-MORL. In Qiao & Wang (2023a), they do not mention explicitly the privacy budgets $\epsilon$, but instead mention the zero-concentrated differential privacy (z-CDP) parameter $\rho$. For clarity and fair comparison, we convert the z-CDP guarantee into a DP guarantee. For this, we use Proposition 1.3 from Bun & Steinke (2016): if a mechanism is $\rho$-zero-concentrated DP, then for any $\delta > 0$ it is $(\epsilon, \delta)$-DP, with $\epsilon = \rho + 2\sqrt{\rho \log(1/\delta)}$. As they evaluate their algorithms for a dataset size up to 1000, we consider two values of $\delta \in \{1/100, 1/1000\}$. Table 5 shows the results for the various parameters $\rho$ mentioned in Figure 1 from Qiao & Wang (2023a). We observe Qiao & Wang (2023a) also considers the low privacy regime with $\rho = 25$ yielding $\epsilon$ close to 50, which is comparable to our low privacy variant. They indeed consider $\epsilon$ close to 1 with $\rho = 0.1$, but the cost is a 2 to 3 times worse utility. Other configurations proposed are closed in privacy budgets to what we consider in our paper. Overall, our work achieves comparable privacy-utility trade-offs than Qiao & Wang (2023a), but on significantly more complex tasks.

Table 5: Results from Qiao & Wang (2023a), converted from z-CDP

| Z-CDP GUARANTEE $\rho$ | DP $\epsilon$ FOR $\delta = 10^{-1}$ | DP $\epsilon$ FOR $\delta = 10^{-3}$ |
|---|---|---|
| 25 | 40.2 | 51.3 |
| 5 | 11.8 | 16.8 |
| 1 | 4.0 | 6.26 |
| 0.1 | 1.1 | 1.8 |

# G DISCUSSION ON THE $\epsilon$ PARAMETER

As the privacy budgets $\epsilon$'s presented in our experimental results do not provide strong theoretical DP guarantees, we would like to further discuss the implications of such privacy budgets in practice.

First, we point out that such $\epsilon$ values are comparable to existing work. In particular, as pointed out in Section F, Qiao & Wang (2023a) achieves similar privacy-performance trade-offs and also consider the "low privacy regime" with $\epsilon$'s approaching 50 for their best-performing variant. We argue that studying different privacy regimes allows us to clearly highlight the trade-offs between privacy and performance.

Moreover, in light of recent literature on achieving differential privacy in practical deep learning (Carlini et al., 2019; Ponomareva et al., 2022; 2023), we argue that these $\epsilon$ values may offer an adequate level of privacy in real-world applications. Ponomareva et al. (2023) states $\epsilon \lessgtr 10$ as a realistic and widely used goal in DP deep learning and a "sweet spot" where it is possible to preserve acceptable utility for complex ML models. Moreover, these studies point out the overly restrictive assumptions on the adversary side, which may yield unnecessarily pessimistic privacy bounds. In

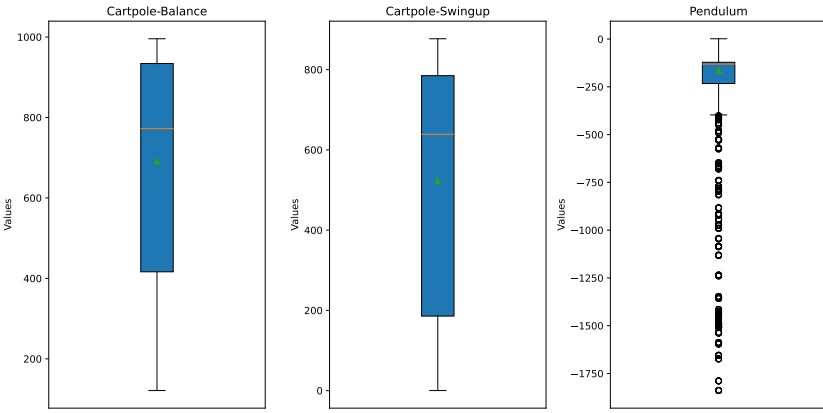

Figure 4: Dataset statistics

offline RL especially, the definition of DP assumes the adversary only has to discriminate between two precise neighboring datasets $D$ and $D' = D \cup \{\tau\}$ as well as the release of all gradients and strong assumptions about the adversary, whereas in practice the adversary faces the much harder task of reconstructing a high-dimensional trajectory based on the output policy and limited side information only.

## H EXPERIMENT DETAILS

### H.1 DATASETS

Table 6 provides additional details on the offline datasets used in experiments. Figure 4 shows episode return statistics for each dataset.

Table 6: Dataset details

|  | CARTPOLE | PENDULUM | HALFCHEETAH |
|---|---|---|---|
| ORIGIN | CUSTOM | CUSTOM | D4RL |
| OBSERVATION SPACE $\mathcal{S}$ | $\mathbb{R}^5$ | $\mathbb{R}^3$ | $\mathbb{R}^{17}$ |
| ACTION SPACE $\mathcal{A}$ | $[-1, 1]$ | $[-2, 2]$ | $[0, 1]^6$ |
| NB. OF EPISODES $K$ | 30,000 | 30,000 | 2,003 |

### H.2 IMPLEMENTATION DETAILS

For all tasks, the model is approximated with a deep neural network with SWISH activation functions and decaying weights. Models take as input a concatenation of the current state $s$ and the taken action $a$ and predict the difference between the next state $s'$ and the current state $s$ along with the reward $r$. Table 7 provides further implementation details.

The code repository for PRIMORL is provided as part of the supplementary material and will be made public upon acceptance. For MOPO, we use the official implementation from `https://github.com/tianheyu927/mopo`, as well as the PyTorch re-implementation from `https://github.com/junming-yang/mopo`. Our implementation of PRIMORL, which mainly uses PyTorch, is also based on these codebases. To collect the datasets, we use DDPG implementation from `https://github.com/schatty/DDPG-pytorch`.

Model training with TDP MODEL ENSEMBLE TRAINING is parallelized over 16 CPUs using JobLib, while SAC training is conducted over a single Nvidia Tesla P100 GPU.

Table 7: Implementation details

|  | CARTPOLE | PENDULUM | HALFCHEETAH |
|---|---|---|---|
| MODEL INPUT DIMENSION | 6 | 4 | 23 |
| MODEL OUTPUT DIMENSION | 6 | 4 | 18 |
| MODEL HIDDEN LAYERS | 2 | 2 | 4 |
| NEURONS PER LAYER | 128 | 64 | 200 |
| WEIGHT DECAY | ✓ | ✓ | ✓ |
| ACTIVATION FUNCTIONS | SWISH | SWISH | SWISH |
| ENSEMBLE SIZE $N$ | 5 | 3 | 7 |

Table 8: Training and Hyperparameters details

|  | CARTPOLE | PENDULUM | HALFCHEETAH |
|---|---|---|---|
| TEST SET SIZE | $1\% \times K$ | $1\% \times K$ | $10\% \times K$ |
| EARLY STOPPING | ✓ PATIENCE = 10 | ✓ PATIENCE = 10 | ✓ PATIENCE = 5 |
| SAMPLING RATIO $q$ | $10^{-3}$ | $10^{-3}$ | $10^{-2}$ |
| MODEL LOCAL EPOCHS $E$ | 1 | 1 | 1 |
| MODEL BATCH SIZE $B$ | 16 | 16 | 16 |
| MODEL LR $\eta$ | $10^{-3}$ | $10^{-3}$ | $10^{-3}$ |
| CLIPPING STRATEGY | FLAT | PER-LAYER | PER-LAYER |
| SAC LR | $3.10^{-4}$ | $3.10^{-4}$ | $3.10^{-4}$ |
| ROLLOUT LENGTH $H$ | 20 | 30 | 5 |
| REWARD PENALTY $\lambda$ | 2.0 | 2.0 | 1.0 |
| AUTO-$\alpha$ | ✓ | ✓ | ✓ |
| TARGET ENTROPY $H$ | -3 | -3 | -3 |
| UNCERTAINTY ESTIMATE | $u_{\text{MPD}}$ (BAL.), $u_{\text{MA}}$ (SWI.) | $u_{\text{MPD}}$ | $u_{\text{MA}}$ |

## H.3 TRAINING DETAILS

Before model training, we split the offline dataset into two parts: a train set used to train the model, and a test set used to track model performance. We consider the test set public so that this operation does not involve additional privacy leakage. The split is made by episode (instead of by transitions), so that the test set contains 1% of the episodes for CARTPOLE and PENDULUM and 20% for HALFCHEETAH. To tune the clipping norm, we set $z = 0$ and progressively decreased $C$ until it started to adversely affect performance provided the best results. Moreover, we set the sampling ratio so that a few dozen episodes are randomly selected at each step, which proved to work best in our experiments, which correspond to $q = 10^{-3}$ for CARTPOLE and PENDULUM. The model is trained until convergence using *early stopping*. Test set prediction error is used to track model improvement. For SAC training, the real-to-model ratio $r_{\text{real}}$ is zero, meaning that SAC is trained using only simulated data from the model, and does not access any data from the offline dataset. Training details are provided in Table 8.

## H.4 HYPERPARAMETERS

The model is trained using TDP MODEL ENSEMBLE TRAINING with learning rate $\eta = 10^{-3}$, batch size $B = 16$, and number of local epochs $E = 1$.

The policy is optimized within the model using Soft Actor-Critic with rollout, with rollout length and penalty depending on the task. We use a learning rate of $3.10^{-4}$ for both the actor and the critic. For entropy regularization, we use auto-$\alpha$ with target entropy $H = -3$.

Hyperparameters are summarized in Table 8. We do not report the privacy loss resulting from hyperparameter tuning, although we recognize its importance in real-world applications.

## H.5 Privacy Parameters

In Table 1, we provide the privacy budgets $\epsilon$ computed with the moments accountant method from Abadi et al. (2016). We use the DP accounting tools from Google's Differential Privacy library, available on GitHub. Privacy budget are computed for $\delta = 10^{-5}$, *i.e.* less than $K^{-1}$ as recommended in the literature. It also depends on the noise multiplier $z$, the number of training round $T$ and the sampling ratio $q$. Since we use early stopping and the different training runs have different durations, we use the average number of training rounds in the privacy budget computations.

For PENDULUM, we use $z = 0.35$ and $z = 0.52$ for PRIMORL LOW and PRIMORL HIGH, respectively. For CARTPOLE-BALANCE, we use $z = 0.25$ and $z = 0.45$ for PRIMORL LOW and PRIMORL HIGH, respectively. For CARTPOLE-SWINGUP, we use $z = 0.25$ and $z = 0.38$ for PRIMORL LOW and PRIMORL HIGH, respectively. The value for PRIMORL HIGH is chosen by incrementally increasing $z$ until policy performance drops below acceptable levels. The corresponding $\epsilon$ is therefore roughly the best privacy budget we can obtain while keeping acceptable policy performance. The value for PRIMORL LOW is chosen arbitrarily to provide a weaker level of privacy that typically yields higher policy performance, illustrating the trade-off between the strength of the privacy guarantee and the performance. Table 9 summarizes the parameters used to compute $\epsilon^{\text{MA}}(z, q, T, \delta)$ in our experiments.

Table 9: Training and privacy parameters used to compute $\epsilon^{\text{MA}}(z, q, T, \delta)$.

|          |      | $z$  | $T$         | $q$       | $\delta$  | $\epsilon$ |
|----------|------|------|-------------|-----------|-----------|------------|
| PENDULUM | LOW  | 0.35 | $7.10^3$    | $10^{-3}$ | $10^{-5}$ | 22.3       |
|          | HIGH | 0.52 | $7.10^3$    | $10^{-3}$ | $10^{-5}$ | **5.1**    |
| BALANCE  | LOW  | 0.25 | $7.10^3$    | $10^{-3}$ | $10^{-5}$ | 85.0       |
|          | HIGH | 0.45 | $7.10^3$    | $10^{-3}$ | $10^{-5}$ | **8.2**    |
| SWINGUP  | LOW  | 0.25 | $10.10^3$   | $10^{-3}$ | $10^{-5}$ | 94.2       |
|          | HIGH | 0.38 | $10.10^3$   | $10^{-3}$ | $10^{-5}$ | **17.0**   |

## H.6 Computing $\epsilon$: The Moments Accountant

Theorem 1 from McMahan et al. (2018) allows us to compute the privacy guarantees $\left(\epsilon^{\text{MA}}(z, q, T, \delta), \delta\right)$ of Algorithm 1 using the Moments Accountant from Abadi et al. (2016). To compute $\epsilon^{\text{MA}}(z, q, T, \delta)$ in our experiments, we use the DP accounting tools from Google's Differential Privacy library, which provides an improved version of the moments accountant based on Rényi Differential Privacy (RDP) Mironov (2017b). Since the computations of the RDP accountant are quite involved while the underlying principles are the same, we rather present the original moments accounting method based on Section 3.2 from Abadi et al. (2016).

By taking into account the DP noise distribution, the moments accountant allows to get a tighter bound on the total privacy leakage compared to the standard strong composition theorem. Using an $(\epsilon, \delta)$-DP mechanism at each gradient step, Algorithm 1 with $T$ training steps and a sampling ratio $q$ is $\left(\mathcal{O}(q\epsilon\sqrt{T}), \delta\right)$-DP by the moments accountant. For comparison, the strong composition theorem would yield $\left(\mathcal{O}(q\epsilon\sqrt{T \log(1/\delta)}), Tq\delta\right)$.

The moments accountant works by computing the log moments of the privacy loss random variable. We denote $\mathcal{M}_{\sigma^2}$ the Gaussian mechanism at each training step $t$, which is characterized by the magnitude $\sigma^2 := \sigma^2(z, q, T, \delta)$ of the Gaussian noise. The privacy loss for $\mathcal{M}_{\sigma^2}$ at output $o$ is defined as follows:

$$c(o; \sigma^2, D, D') = \log \frac{\mathbb{P}(\mathcal{M}_{\sigma^2}(D) = o)}{\mathbb{P}(\mathcal{M}_{\sigma^2}(D') = o)} \ ,$$

where $D$, $D'$ are neighboring datasets. It quantifies the privacy leakage for the specific output $o$ taking into account the randomness of the algorithm. The $\lambda$-th moment $\alpha(\lambda; D, D')$ is defined as the

logarithm of the moment generating function:

$$\alpha_{\mathcal{M}_{\sigma^2}}(\lambda) = \max_{D,D'} \log \mathbb{E}_{o \sim \mathcal{M}_{\sigma^2}(D)} \left[ \exp(\lambda c(o; \mathcal{M}_{\sigma^2}, D, D')) \right] \ .$$

To bound $\alpha_{\mathcal{M}_{\sigma^2}}(\lambda)$ for a Gaussian mechanism of scale $\sigma^2$, Abadi et al. (2016) show that, denoting $\mu_x$ the p.d.f. of $\mathcal{N}(x, \sigma^2)$ and $\mu = (1-q)\mu_0 + q\mu_1$, it suffices to estimate $\alpha(\lambda) = \log \max(E_1, E_2)$ with:

$$E_1 = \mathbb{E}_{z \sim \mu_0} \left[ (\mu_0(z)/\mu(z))^\lambda \right]$$
$$E_2 = \mathbb{E}_{z \sim \mu} \left[ (\mu(z)/\mu_0(z))^\lambda \right] \ .$$

Implementations of the moments accountant typically use numerical integration to estimate $\alpha(\lambda)$.

To compute $\epsilon^{\text{MA}}(z, q, T, \delta)$, a bound on the total privacy loss of Algorithm 1, it then suffices to compute a bound on $\alpha_{\mathcal{M}_{\sigma^2}}(\lambda)$ at each step and sum over all steps. Since we cannot compute a bound for all $\lambda$, we need to specify as input a discrete list $\Lambda = \{\lambda_1, ..., \lambda_S\}$ of moments to bound, and select the $\lambda$ yielding the best privacy budget. Abadi et al. (2016) find that it usually suffices to compute $\alpha(\lambda)$ for $\lambda \leq 32$ (see Section 4).

### H.7 COMPUTATIONAL RESOURCES

We perform training on a single machine with 64 CPUs and 6 Tesla P100 GPUs with 16GB RAM each. The full training of a single policy, from model learning to policy optimization, takes several hours.

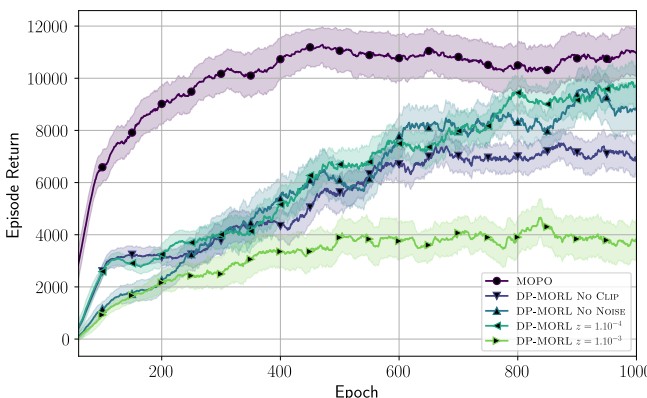

Figure 5: Learning curves for the SAC policy on HALFCHEETAH (*right*). Policy performance (episodic return) is evaluated in the true MDP at the end of each training epoch, over 10 evaluation episodes with different random seeds.

## I  ADDITIONAL EXPERIMENTS

Table 10: Ablation study investigating the impact of using different clipping methods (*flat clipping* (FC) and *per-layer clipping* (LC)) and different uncertainty estimates ($u_{MA}$ and $u_{MPD}$). For instance, MA + LC means the model has been train with *per-layer clipping* and the policy optimized under uncertainty estimate $u_{MA}$. We report policy performance as the mean episodic return over 10 evaluation episodes, averaged over the last 10 epochs of policy optimization. Average performance and 95% confidence intervals are computed on at least 3 seeds.

|          | z    | MA + LC          | MPD + LC          | MA + FC          | MPD + FC           |
|----------|------|------------------|-------------------|------------------|--------------------|
| PENDULUM | 0.35 | $734.5 \pm 28.9$  | $\mathbf{817.4 \pm 21.7}$ | $638.9 \pm 91.7$  | $757.9 \pm 76.5$   |
|          | 0.52 | $750.1 \pm 75.97$ | $\mathbf{778.9 \pm 53.5}$ | $612.4 \pm 76.8$  | $723.0 \pm 47.0$   |
| BALANCE  | 0.25 | $785.4 \pm 77.9$  | $792 \pm 85.8$    | $\mathbf{819.9 \pm 66.1}$ | $815.8 \pm 97.2$   |
|          | 0.45 | $749.0 \pm 115.6$ | $738.5 \pm 85.7$  | $722.7 \pm 105.7$ | $\mathbf{758.2 \pm 187.2}$ |
| SWINGUP  | 0.25 | $711.4 \pm 90.6$  | $704.2 \pm 46.2$  | $772.4 \pm 73.9$  | $\mathbf{777.1 \pm 35.0}$  |
|          | 0.38 | $536.1 \pm 112.8$ | $575.4 \pm 89.7$  | $\mathbf{698.3 \pm 57.5}$ | $590.8 \pm 34.8$   |

In Table 10, we compare performance depending on the clipping method used to train the model (*flat or per-layer clipping*) and the uncertainty estimator used to optimize the policy ($u_{MPD}$ or $u_{MA}$). We can see that no clipping method or uncertainty estimator is significantly superior overall, but these choices may impact privacy performance for a specific task. Preliminary results on this ablation study led us to choose, for each task, the clipping strategy and the uncertainty estimate stated in Table 8.

## J  EXPERIMENTS ON HALFCHEETAH

We conduct experiments on the MEDIUM-EXPERT dataset ($K = 2,003$) from the classic D4RL benchmark (Fu et al., 2020). Experimental results are reported in Figure 5 and Table 11 (in appendix), using $C = 15.0$ and $q = 10^{-2}$.

If PRIMORL can train competitive policies with small enough noise levels — a tiny amount of noise like $z = 10^{-4}$ proving even beneficial, possibly acting as a kind of regularization —, we were not able to obtain reasonable $\epsilon$'s. Indeed, a noise multiplier as small as $z = 10^{-3}$ is enough to cause a significant decline in performance. HALFCHEETAH thus appears a significantly harder tasks than CARTPOLE and PENDULUM. It is not surprising as HALFCHEETAH is higher-dimensional, and the theoretical analysis led in Section 4.3.1 showed that the dimension $d$ of the problem could negatively impact the performance of the policy. However, we point out that the size of the dataset for HALFCHEETAH is very limited, and argue that larger datasets with substantially more episodes would translate into competitive privacy-performance trade-offs, as we develop in Section L.

Table 11: Results for HALFCHEETAH MEDIUM-EXPERT. RETURN is the return of the SAC policy evaluated over 10 episodes at the end of each training epoch, averaged over the last 20 epochs.

| METHOD | $z$ | RETURN |
|---|---|---|
| MOPO | 0.0 | $10931 \pm 1326$ |
| PRIMORL NO CLIP | 0.0 | $7062 \pm 2230$ |
| PRIMORL NO NOISE | 0.0 | $8792 \pm 2053$ |
| PRIMORL | $z = 1.10^{-4}$ | $9729 \pm 2018$ |
|  | $z = 1.10^{-3}$ | $3697 \pm 1465$ |

## K   ALGORITHMS

Algorithm 2 is the fully detailed pseudo-code for PRIMORL. Algorithm 3 details the clipping method used in TDP MODEL ENSEMBLE TRAINING. Algorithm 4 is the pseudo-code for SAC policy optimization on the pessimistic private model. This pseudo-code is based on `https://spinningup.openai.com/en/latest/algorithms/sac.html`

---

**Algorithm 2** Model Training with TDP MODEL ENSEMBLE TRAINING

1: **Input:** offline dataset $\mathcal{D}_K$, sampling ratio $q \in (0,1)$, noise multiplier $z \geq 0$, clipping norm $C > 0$, local epochs $E$, batch size $B$, learning rate $\eta$
2: **Output:** private model $\hat{M}_\theta$
3: Initialize model parameters $\theta_0$
4: **for** each iteration $t \in [\![0, T-1]\!]$ **do**
5: $\quad \mathcal{U}_t \leftarrow$ (sample with replacement trajectories from $\mathcal{D}_K$ with prob. $q$)
6: $\quad$ **for** each trajectory $\tau_k \in \mathcal{U}_t$ **do**
7: $\quad\quad$ Clone current models $\{\theta_i^{\text{start}}\}_{i=1}^N \leftarrow \{\theta_i(t)\}_{i=1}^N$
8: $\quad\quad \theta \leftarrow \theta^{\text{start}} := (\theta^{\text{start}})_{i=1}^N$
9: $\quad\quad$ **for** each local epoch $i \in [\![1, E]\!]$ **do**
10: $\quad\quad\quad \mathcal{B} \leftarrow$ ($\tau_k$'s data split into size $B$ batches)
11: $\quad\quad\quad$ **for** each batch $b \in \mathcal{B}$ **do**
12: $\quad\quad\quad\quad \theta \leftarrow \theta - \eta \nabla \mathcal{L}(\theta; b)$
13: $\quad\quad\quad\quad \theta \leftarrow \theta^{\text{start}} + \text{ENSEMBLECLIP}(\theta - \theta^{\text{start}}; C)$
14: $\quad\quad\quad$ **end for**
15: $\quad\quad$ **end for**
16: $\quad\quad \Delta_{t,k}^{\text{clipped}} \leftarrow \theta - \theta^{\text{start}}$
17: $\quad$ **end for**  $\left. \right\}$ ENSCLIPGD $\left( \tau_k, \{\theta_i^{\text{start}}\}_{i=1}^N ; C, E, B \right)$
18: $\quad \Delta_i^{\text{avg}}(t) = \frac{\sum_{k \in \mathcal{U}_t} \Delta_{i,k}^{\text{clipped}}(t)}{qK}$
19: $\quad \theta(t+1) \leftarrow \theta(t) + \Delta^{\text{avg}}(t) + \mathcal{N}\left(0_{N_d}, \left(\frac{zC}{qK}\right)^2 I_{N_d}\right)$
20: **end for**

---

**Algorithm 3** Ensemble Clipping (ENSEMBLECLIP)

1: **Input:** ensemble size $N$, number of model layers $L$, unclipped gradient $\Delta = \{\Delta_{i,\ell}\}_{i,\ell=1}^{N,L}$, clipping norm $C$
2: **Output:** clipped gradient $\Delta^{\text{clipped}}$
3: $\Delta_i \leftarrow (\Delta_{i,\ell})_{\ell=1}^L, \;\; C_i = \frac{C}{\sqrt{N}}$
4:
$$\Delta_i^{\text{clipped}} \leftarrow \frac{\Delta_i}{\max\left(1, \frac{\|\Delta_i\|_2}{C_i}\right)}, \;\; j = 1, ..., m.$$

---

**Algorithm 4** Private Model-Based Optimization with SAC

---

1: **Input:** private model $\hat{M} = (\hat{P}, \hat{r})$, empty replay buffer $\mathcal{B}$, uncertainty estimator $u \in \{u_{\text{MA}}, u_{\text{MPD}}\}$
2: **Output:** private policy $\hat{\pi}^{\text{DP}}$
3: Initialize policy parameters $\xi$, Q-function parameters $\omega_1, \omega_2$ and target parameters $\omega_{\text{targ},1}, \omega_{\text{targ},2}$
4: **for** epoch $e \in [\![1, E]\!]$ **do**
5:   **while** episode is not terminated **do**
6:     Observe state $s$ and select action $a \sim \pi_\xi(\cdot|s)$
7:     Execute $a$ in the pessimistic MDP $\tilde{\mathcal{M}}$ and observe next state $s' \sim \hat{P}(\cdot|s, a)$, reward $r \sim \hat{r}(s, a) - \lambda u(s, a)$ and done signal $d$
8:     Store $(s, a, r, s', d)$ in replay buffer $\mathcal{B}$
9:     **if** time to update **then**
10:       Sample a batch of transitions $B = \{(s, a, r, s', d)\}$ from buffer $\mathcal{B}$
11:       Compute targets for Q-functions:

$$y(r, s', d) = r + \gamma(1 - d)\left(\min_{i=1,2} Q_{\omega_{\text{targ},i}}(s', \tilde{a}') - \alpha \log \pi_\xi(\tilde{a}'|s')\right), \ \ \tilde{a}' \sim \pi_\xi(\cdot|s') \ .$$

12:       Update Q-functions by one step of gradient descent using:

$$\nabla_{\omega_i} \frac{1}{|B|} \sum_{(s,a,r,s',d) \in B} (Q_{\omega_i}(s, a) - y(r, s', d))^2, \ \ \text{for } i = 1, 2.$$

13:       Update policy by one step of gradient ascent using:

$$\nabla_\xi \frac{1}{|B|} \sum_{s \in B} \left(\min_{i=1,2} Q_{\omega_i}(s, \tilde{a}_\zeta(s)) - \alpha \log \pi_\xi(\tilde{a}_\zeta(s)|s)\right), \ \ \tilde{a}_\zeta(s) \sim \pi_\xi(\cdot|s).$$

14:       Update target networks with:

$$\omega_{\text{targ},i} \leftarrow \rho\omega_{\text{targ},i} + (1 - \rho)\omega_i, \ \ \text{for } i = 1, 2 \ .$$

15:     **end if**
16:   **end while**
17:   Evaluate $\pi_\xi$ is the true environment $\mathcal{M}$.
18: **end for**

---

## L    THE PRICE OF PRIVACY IN OFFLINE RL

In this section, we provide theoretical and practical arguments to further justify the need for (much) larger datasets in order to achieve competitive privacy trade-offs in offline RL, as pointed out in (Section 5).

**Why does privacy benefit so much from large datasets?** From a theoretical perspective, it stems from two facts: 1) $\epsilon$ scales with the sampling ratio $q$ (*privacy amplification by subsampling*), and 2) noise magnitude $\sigma$ is inversely proportional to $\mathbb{E}\left[|\mathcal{U}_t|\right] = qK$. Clearly, the privacy-performance trade-off would benefit from both small $q$ (reducing $\epsilon$) and large $qK$ (reducing noise levels and thus improving performance), which are conflicting objectives for a fixed $K$. However, if we consider using larger datasets of size $K' \gg K$, it becomes possible to find a $K'$ large enough so that we can use $q' \ll q$ and $q'K' \gg qK$, achieving both much stronger privacy and better performance. We can even argue that for a given privacy budget $\epsilon$ (obtained for a given $q$) and an unlimited capacity to increase $K$, we could virtually tend to zero noise levels and achieve optimal performance. Therefore, PRIMORL, already capable of producing good policies with significant noise levels and $\epsilon$, has the potential to achieve stronger privacy guarantees provided access to large enough datasets.

An aspect that deserves further development is the iterative aspect of the used training methods and its effect on privacy. Differential privacy being a worst-case definition, it assumes that all intermediate models are released during training. Although the practicality of this hypothesis is debatable, it definitely impacts privacy: privacy loss is incurred at each training iteration (corresponding to a gradient step on the global model in DP-SGD and TDP MODEL ENSEMBLE TRAINING) and privacy budget, therefore, scales with the number of iterations $T$. Consequently, limiting the number of iterations is even more crucial with DP training than with non-private training. Training a model on the kind of tasks we considered nonetheless requires a lot of iterations to reach convergence (empirically, thousands of iterations for CARTPOLE and tens of thousands of iterations for HALFCHEETAH), and the privacy budget suffers unavoidably.

However, one way to circumvent this is to leverage privacy amplification by subsampling. Indeed, as McMahan et al. (2017) observe, the additional privacy loss incurred by additional training iterations becomes negligible when the sampling ratio $q$ is small enough, which is a direct effect of privacy amplification by subsampling. We discussed above how increasing dataset size $K$ allowed to decrease both sampling ratio $q$ and noise levels. Therefore, by increasing the size of the dataset, we also greatly reduce the impact of the number of training iterations, likely promoting model convergence. This further reinforces the need for large datasets in offline RL in order to study privacy. As an example, McMahan et al. (2018) consider datasets with $10^6$ to $10^9$ users to train DP recurrent language models, and this is arguably the main reason why they achieve formal strong privacy guarantees. For comparison, the classical RL UNPLUGGED and D4RL benchmarks provide datasets with $K \approx 10^1$ to $K \approx 10^3$ datasets. Achieving the privacy-performance trade-offs demonstrated in Section 5 would not have been possible without the collection of large datasets. Moreover, datasets orders of magnitude larger would be required to attain formal, strong privacy guarantees, such as $\epsilon < 1$. While conducting experiments in deep offline RL with such extensive datasets demands substantial computational resources, we argue that scenarios involving access to datasets with a vast number of trajectories are reflective of real-world situations. For this reason, we consider this case worthy of thorough investigation.

Figure 6 illustrates this point in another way. Given $\epsilon \in \{10^{-4}, 10^{-3}, 10^{-2}\}$, we plot for a range of sampling ratio $q$ the maximum number of iterations $T$ that is allowed so that the total privacy loss does not exceed $\epsilon$, as a function of the noise multiplier $z$. We can see how decreasing $q$ makes it well easier to train a private model: dividing $q$ by 10, we "gain" roughly 10 times more iterations across all noise levels.

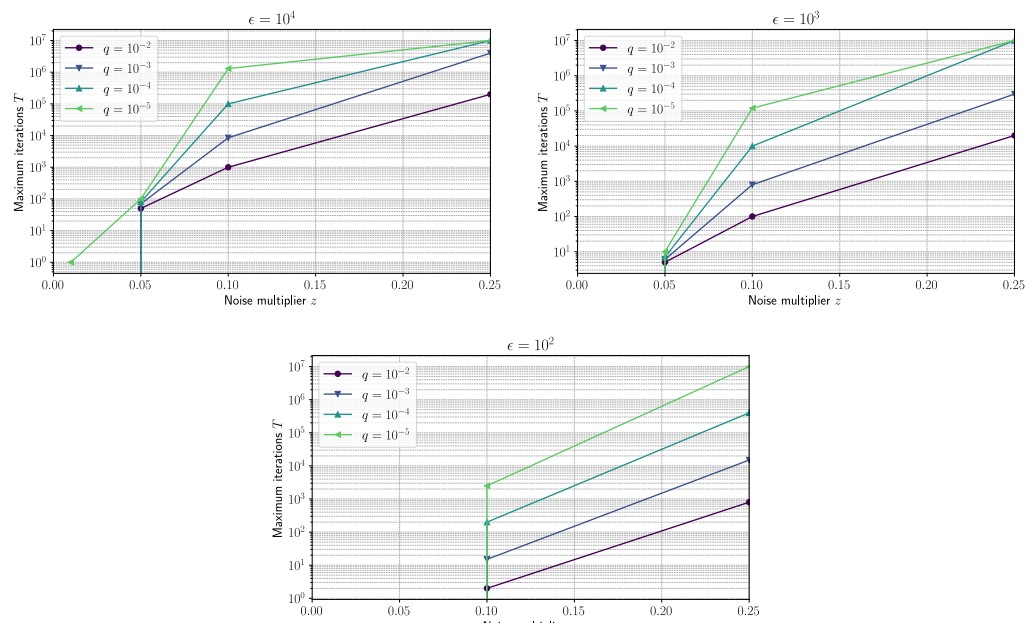

Figure 6: Maximum number of iterations $T$ so that the privacy loss does not exceed $\epsilon$, as function of the noise multiplier $z$.

## M BROADER IMPACTS

As recent advances in the field have moved reinforcement learning closer to widespread real-world application, from healthcare to autonomous driving, and as many works have shown that it is no more immune to privacy attacks than any other area in machine learning, it has become crucial to design algorithmic techniques that protect user privacy. In this paper, we contribute to this endeavor by introducing a new approach to privacy in offline RL, tackling more complex control problems and thus paving the way towards real-world private reinforcement learning. We firmly believe in the importance of pushing the boundaries of this research field and are hopeful that this work will contribute to practical advancements in achieving trustworthy machine learning.

