# OpenReview forum: "Differentially Private Deep Model-Based Reinforcement Learning"
_ICLR.cc/2025/Conference — Submitted to ICLR 2025_

### Official Review · Reviewer_wk8A · 2024-10-25

**Soundness:** 3
**Presentation:** 3
**Contribution:** 2
**Rating:** 6
**Confidence:** 4

**Summary:**

In this paper, the authors consider private deep offline reinforcement learning (RL), where the goal is to train a policy on standard control tasks that is differentially private (DP) with respect to individual trajectories in the dataset. To achieve this, they introduce PriMORL, a model-based RL algorithm with formal differential privacy guarantees. PriMORL first learns an ensemble of trajectory-level DP models of the environment from offline data. It then optimizes a policy on the penalized private model, without any further interaction with the system or access to the dataset. In addition to theoretical guarantees, they empirically demonstrate that PriMORL enables the training of private RL agents on offline continuous control tasks with deep function approximations.

**Strengths:**

1. The problem of offline RL with DP is important and well-motivated.
2. This paper proposes a practical solution to the problem.
3. The authors do experiments to support their algorithm.
4. The paper is well-written in general.

**Weaknesses:**

1. The main concern is about technical novelty. The definition of Trajectory-level DP is directly adapted from [1]. The first part directly applies DP-FEDAVG, while the second part is about learning from the private model with pessimism. To the best of my knowledge, [1] is based on the same idea of private model + pessimism. The DP guarantee for the private model is from previous results, and the DP guarantee for learning the policy is from standard post-processing. I do not see any technical challenge in the process. It will be better if the authors could discuss about the challenges.

[1] Dan Qiao and Yu-Xiang Wang. Offline reinforcement learning with differential privacy.

2. Proposition 4.4 only provides an error bound for estimating the value function of $\hat{\pi}$, which is not standard. Is it possible to derive any results about the sub-optimality gap $V^\star-V^{\hat{\pi}}$?

3. In the experiments, the privacy protection is very weak (some $\epsilon$ being close to 100). What will happen for more practical choices of $\epsilon$? E.g. $\epsilon \approx 1$.

**Questions:**

Please refer to the weaknesses above.

---

> ### Author Response · Authors · 2024-11-18
> **Same Review as NeurIPS**
>
> This review is exactly the same as the one we received for NeurIPS. We have since made many modifications to the paper. In particular, motivated by your feedback, we have developed the theoretical analysis of the different components of our algorithm and provided thorough discussions about the challenges. Also, the private training algorithm has been improved, yielding stronger experimental results. Since this is a new paper, we think a new review is needed.
>
> In the following, we nonetheless carefully address your concerns.

---

> ### Author Response · Authors · 2024-11-18
> **Rebuttal 1/4**
>
> ``The main concern is about technical novelty. The definition of Trajectory-level DP is directly adapted from [1]. The first part directly applies DP-FEDAVG, while the second part is about learning from the private model with pessimism. To the best of my knowledge, [1] is based on the same idea of private model + pessimism. The DP guarantee for the private model is from previous results, and the DP guarantee for learning the policy is from standard post-processing. I do not see any technical challenge in the process. It will be better if the authors could discuss about the challenges.``
>
> [1] indeed uses the idea of training a private model, but only in the context of tabular and linear MDPs. Moreover, experiments in [1] are limited to simple numerical simulations. In this work, we tackle more complex tasks in the general function approximation setting, especially neural network approximations, which introduces very different challenges that have not yet been addressed in the literature.
>
> In the paper, we thoroughly discuss the challenges of obtaining trajectory-level DP in the context of model ensembles, where a direct use of DP-SGD would be ineffective (Section 4.2). We identified that the idea behind DP-FEDAVG, which had not been used in the context of RL, could be effectively used in our setting, and we carefully ensure that the resulting model is private by distributing the gradient clipping across the $N$ models of the ensemble, before discussing how this may impact model convergence in practice. In the following (section 4.3), we analyze theoretically the impact of private model training on policy optimization. Therefore, neither model training nor privacy guarantees is straightforward (and the DP guarantees for the policy are always obtained from standard post-processing).
>
> In our experiments, we are also the first to address more challenging RL tasks with deep neural approximations and believe that our experimental results are overall much stronger than those from concurrent work in [1]. Indeed, our work achieves comparable privacy-utility trade-offs than [1], but on significantly more complex tasks, which we demonstrate in the following. We also updated the paper to enable a clearer comparison with [1] (see experimental section and Section F of the appendix).
>
> [1] evaluate their algorithms on an episodic synthetic linear MDP with 2 states and 100 actions, and horizon $H=20$. In their results, they do not mention explicitly the privacy budgets $\epsilon$, but instead mention the zero-concentrated differential privacy (z-CDP) parameter $\rho$. For clarity and fair comparison, we convert the z-CDP guarantee into a DP guarantee. For this, we use Proposition 1.3 from [3]: if a mechanism is $\rho$-z-CDP, then for any $\delta > 0$ it is $(\epsilon, \delta)$-DP, with $\epsilon = \rho + 2 \sqrt{\rho \log(1 / \delta)}$. As they evaluate their algorithms for a dataset size up to 1000, we consider two values of $\delta \in \{1/100, 1/1000\}$. The table below shows the results for the various parameters $\rho$ mentioned in Figure 1 from [1].
>
> | $\rho$ | $\epsilon$ for $\delta=10^{-1}$ | $\epsilon$ for $\delta=10^{-3}$ |
> |--------|----------------------------------|----------------------------------|
> | 25     | 40.2                             | 51.3                             |
> | 5      | 11.8                             | 16.8                             |
> | 1      | 4.0                              | 6.26                             |
> | 0.1    | 1.1                              | 1.8                              |
>
> Therefore, [1] also considers the low privacy regime with $\rho=25$ yielding $\epsilon$ close to 50, which is comparable to our low privacy variant. They indeed consider $\epsilon$ close to 1 with $\rho=0.1$, but the cost is a 2 to 3 times worse utility on this simple MDP. Other proposed configurations are close in privacy budgets to what we consider in our paper. Overall, our work achieves comparable privacy-utility trade-offs than [1], but on significantly more complex tasks.

---

> ### Author Response · Authors · 2024-11-18
> **Rebuttal 2/4**
>
> ``Proposition 4.4 only provides an error bound for estimating the value function of $\hat{\pi}$ which is not standard. Is it possible to derive any results about the sub-optimality gap $V^\star - V^{\hat{\pi}}$?``
>
> Thank you for the interesting observation. In this work, we disrupt the model convergence with private training, which directly impacts the valuation gap $\vert V^{\hat{\pi}} - \hat{V}^{\hat{\pi}} \vert$ quantifying the divergence of the value of the learned policy between the true and the estimated MDP. We therefore found particularly interesting to quantify how correctly the learned policy was evaluated under the private model, especially since we use the model as a simulator to optimize the policy without any further interactions with the environment. The underlying idea is that we want to control how privacy degrades the quality of the simulator.
>
> However, we agree that a bound on the sub-optimality gap is worthwhile to quantify the end performance of the policy, and observe that the results from propositions 4.3 and 4.4 actually hold for any policy $\pi$. Therefore, we can use the following to derive a bound on the sub-optimality gap based on the valuation gap:
> $$
> V^\star - V^{\hat{\pi}}
> = V^\star - \hat{V}^{\pi^\star} + \hat{V}^{\pi^\star} - \hat{V}^{\hat{\pi}} + \hat{V}^{\hat{\pi}} - V^{\hat{\pi}} \\
> \le (V^\star - \hat{V}^{\pi^\star}) + 0 + (\hat{V}^{\hat{\pi}} - V^{\hat{\pi}}) \\
> \le \sup_{\pi \in \Pi} \vert V^\pi - \hat{V}^\pi \vert + \sup_{\pi \in \Pi} \vert V^\pi - \hat{V}^\pi \vert \\
> = 2 \sup_{\pi \in \Pi} \vert V^\pi - \hat{V}^\pi \vert \enspace,
> $$
>
> where the first inequality comes from the fact that $\hat{\pi} \in \text{arg}\max_{\pi \in \Pi} \hat{V}^\pi$, and the second inequality is due to each of the terms being the value gap between the true and the estimated dynamics for a fixed policy ($\pi^\star$ and $\hat{\pi}$, respectively).
>
> We have clarified this point and added the above result in the updated version of the paper.

---

> ### Author Response · Authors · 2024-11-18
> **Rebuttal 3/4**
>
> ``In the experiments, the privacy protection is very weak (some $\epsilon$ being close to 100). What will happen for more practical choices of $\epsilon$ ? E.g. $\epsilon \approx 1$``
>
> We acknowledge your concerns over the level of privacy protection, but these values must be appreciated within the entire context of the paper and literature. Here, we propose a framework with two distinct levels of privacy, where larger values of $\epsilon$ explicitly correspond to "low privacy regime". This allows us to clearly highlight the trade-offs between privacy and performance. As pointed out above, previous work [1] also consider a similar low privacy regime in their experiments.
>
> With smaller $\epsilon$ values corresponding to a higher privacy regime (within the 5 to 20 range), our algorithm retains good performance. We do observe more substantial degradation as $\epsilon$ approaches 1, as shown in Figure 4, but not unlike previous algorithms from [1] which also suffers significant performance degradation for $\epsilon$ close to 1, although on much simpler tasks. Overall, our work achieves comparable privacy-utility trade-offs than [1], but on significantly more complex tasks.
>
> Moreover, we must reconsider the context of achieving privacy in deep RL, and we have thoroughly discussed the implications of our empirical results in Sections 5.2 and 6, as well as in Appendix J. In light of recent literature on achieving practical privacy in deep learning, we argue that these $\epsilon$ values may offer an adequate level of privacy in real-world applications. [2] states $\epsilon \lessapprox 10$ as a realistic and widely used goal in DP deep learning and a "sweet spot" where it is possible to preserve acceptable utility for complex ML models. Additionally, we outline approaches to achieve better trade-offs between privacy and performance.
>
> [1] Dan Qiao and Yu-Xiang Wang. Offline reinforcement learning with differential privacy.
>
> [2] Ponomareva et al. How to DP-fy ML: A Practical Guide to Machine Learning with Differential Privacy. 2023.
>
> [3] Bun and Steinke, Concentrated Differential Privacy: Simplifications, Extensions, and Lower Bounds.

---

> ### Author Response · Authors · 2024-11-18
> **Rebuttal 4/4**
>
> Given that we have addressed all the comments from your review in this new version of the paper, and have also included additional remarks to enable a comparison with [1] in the last revision, would you consider increasing your score?

---

> ### Author Response · Authors · 2024-11-22
>
> Dear Reviewer,
>
> We wanted to check if you have had the opportunity to consider the new version of our paper and the points discussed above. Should you have any further questions or need clarification on any of the matters you raised, we would be glad to engage in further discussion.

---

> > ### Comment · Reviewer_wk8A · 2024-11-23
> >
> > Thanks for your detailed explanation.
> >
> > The explanation about the experiment well addressed my concern. I believe your approach achieves at least comparable performance to [1] on a more general setting. In addition, the second point demonstrates the closeness of two models, which is a nice theoretical statement. I understand it is challenging to derive some sub-optimality bound for the final output policy.
> >
> > Overall, I appreciate your effort and would like to raise my score.

---

> > > ### Author Response · Authors · 2024-11-25
> > >
> > > Thank you for your time and positive feedback. We are happy that this discussion has effectively addressed your concerns.

---

### Official Review · Reviewer_h3Xi · 2024-10-29

**Soundness:** 2
**Presentation:** 3
**Contribution:** 2
**Rating:** 6
**Confidence:** 4

**Summary:**

This paper introduces PRIMORL, a differentially private (DP) model-based reinforcement learning algorithm for offline settings. PRIMORL trains policies for continuous control tasks while ensuring trajectory-level privacy by learning an ensemble of DP models from offline data. This approach protects against privacy leaks in RL, especially critical in applications where individual trajectories may contain sensitive information. PRIMORL operates in an offline, infinite-horizon setting, leveraging private models to optimize policies without further interaction with the environment. Empirically, PRIMORL demonstrates competitive performance on deep RL tasks, advancing private RL beyond simpler tabular and linear MDPs and addressing practical privacy-performance trade-offs.

**Strengths:**

The paper is clearly organized. The authors first introduce the trajectory-level differential privacy framework within the offline RL setting, then explain the method for training private models using a model ensemble approach, covering both implementation details and theoretical guarantees. Finally, they describe how policy optimization is achieved by incorporating uncertainty techniques, with theoretical support provided as well.

**Weaknesses:**

1.	It is unclear how the private-variant experiments control for \epsilon. In the theoretical guarantees section, \epsilon is presented as a theoretical bound, yet here it seems to be treated as a tunable hyperparameter, with little explanation connecting these two perspectives.
2.	While the motivation for the work is compelling, the experimental design is relatively simple and basic. I would have liked to see experiments that address the unique challenges of applying DP frameworks within the RL domain, yet this paper lacks a broader experimental analysis to underscore the real-world relevance of introducing DP into RL.

**Questions:**

1.In Theorem 4.2, a general formula for \epsilon^{MA}(z, q, T, \delta) is presented, but neither the main text nor the appendix provides a complete, detailed expression of this formula. Could you include a full derivation of this formula so we can clearly understand how \epsilon^{MA} is calculated based on inputs like z, q, T, and \delta?
2.In Section 4.3.2, the paper discusses handling model uncertainty under a private setting but appears to apply existing uncertainty-handling techniques from non-private settings directly to the private setting. Could you clarify any special considerations or unique aspects of handling uncertainty in the private setting? Specifically, how might model error and uncertainty differ under a private setting, given its unique constraints? We would appreciate any further insights on this point.

---

> ### Author Response · Authors · 2024-11-18
> **Rebuttal 1/3**
>
> Thank you for the positive comments and the constructive feedback. In the following, we carefully address your concerns.
>
> ``It is unclear how the private-variant experiments control for $\epsilon$. In the theoretical guarantees section, $\epsilon$ is presented as a theoretical bound, yet here it seems to be treated as a tunable hyperparameter, with little explanation connecting these two perspectives.} AND \textbf{In Theorem 4.2, a general formula for $\epsilon^{MA}(z, q, T, \delta)$ is presented, but neither the main text nor the appendix provides a complete, detailed expression of this formula. Could you include a full derivation of this formula so we can clearly understand how $\epsilon^{MA}$ is calculated based on inputs like $z, q, T$, and $\delta$?``
>
> Thank you for your comments regarding the computation of the privacy budget $\epsilon$ in our experiments, and how it relates to the perspective of $\epsilon$ as a theoretical upper bound. In the following, we would like to clarify these points.
>
> $\epsilon$ indeed represents a theoretical upper bound on the privacy leakage of the training mechanism and could then, at first, be interpreted as a parameter of the problem. However, in practice, $\epsilon$ depends on the scale $\sigma^2$ of the Gaussian noise added to the gradients: a larger perturbation $\sigma^2$ implies a stronger privacy and a smaller value of $\epsilon$. In our private training method, $\sigma^2$ depends directly on the tunable hyperparameters $z, q, T$ and $\delta$. Therefore, $\epsilon$ is not a hyperparameter itself, but is computed a posteriori based on the values of the above hyperparameters.
>
> In our experiments, we fix $q, T$ and $\delta$ and tune the noise multiplier $z$ to make $\epsilon$ vary (the higher $z$, the smaller $\epsilon$), in order to showcase the performance of our algorithm under various levels of privacy, which is a standard evaluation protocol for DP algorithms. This eventually demonstrates that our method is capable of achieving good privacy-performance trade-offs. We point out that we can only compute a theoretical upper bound on the privacy leakage. There is no standardized approach to evaluate $\epsilon$ empirically, and empirical measures would only yield a lower bound which can be far from the true privacy leakage.
>
> There are different ways of computing an $\epsilon$ bound, all using DP composition properties, and we want to compute the tightest possible bound to obtain the best possible privacy-utility trade-off for our algorithm. In our work, given hyperparameters $z, q, T, \delta$, $\epsilon^{MA}(z, q, T, \delta)$ is the $\epsilon$ value computed by the moments accountant from [1], as we state in Theorem 4.2. Unfortunately, there is no simple explicit formula for it: at each training step $t$, denoting $\mathcal{M}_t$ the Gaussian mechanism outputting the noisy gradient, the moments accountant computes a bound on the log moments of the privacy loss random variable $\log \frac{\mathcal{M}_t(d) = o}{\mathcal{M}_t(d) = o}$, using numerical integration ($d$ and $d^\prime$ are neighboring datasets, and $o$ is a given output). The procedure is detailed in Section 3.2 from [1], and we use the implementation from the DP accounting tools from Google’s Differential Privacy library which provides an improved version of the moments accountant based on the Rényi divergence. We have added an additional section (F.6) in the appendix to present the moments accountant more in details in the revised version of our paper.
>
> [1] Deep Learning with Differential Privacy, Abadi et al., 2016.

---

> ### Author Response · Authors · 2024-11-18
> **Rebuttal 2/3**
>
> ``While the motivation for the work is compelling, the experimental design is relatively simple and basic. I would have liked to see experiments that address the unique challenges of applying DP frameworks within the RL domain, yet this paper lacks a broader experimental analysis to underscore the real-world relevance of introducing DP into RL.``
>
> As we point out in the introduction and related work sections, RL is already used in personalized services on sensitive data, and has been shown to be vulnerable to privacy attacks, especially membership inference attacks (see [2]). We believe these are sufficient reasons in themselves to introduce DP in RL. We agree that it would be worthwhile to lead experiments based on real-world data, but this seems unrealistic at the moment. Moreover, our experimental design is in line with standard empirical practices in RL, and the empirical evaluation of our method is arguably stronger than existing works in DP RL. We would be grateful if you could elaborate on the experiments you would have liked to see in our paper, in order to better address your concerns and improve both the current study and future research.
>
> [2] Gomrokchi et al. Membership Inference Attacks Against Temporally Correlated Data in Deep Reinforcement Learning.
>
> ``In Section 4.3.2, the paper discusses handling model uncertainty under a private setting but appears to apply existing uncertainty-handling techniques from non-private settings directly to the private setting. Could you clarify any special considerations or unique aspects of handling uncertainty in the private setting? Specifically, how might model error and uncertainty differ under a private setting, given its unique constraints? We would appreciate any further insights on this point.``
>
> We indeed apply existing uncertainty-handling techniques from the non-private offline MBRL literature (reward penalization with a measure of model uncertainty). We justify this choice based on the theoretical analysis led in Section 4.3.1 and related work discussion from 4.3.2. This choice is further supported by the empirical evaluation of our method.
>
> The theoretical analysis from Section 4.3.1 shows how and why private model training impacts the reliability of our model for evaluating policies, which can lead to misjudging the quality of a policy in the true environment. Therefore, we must carefully consider the implications in terms of how uncertainty is handled in the private setting. The simulation lemma suggests that the quality of the model as a simulator can be impacted by both the model error and the distribution shift; however, in the updated version of the paper, we show that private training only increases model error (scaling with $1/\sqrt{\epsilon}$), and not distribution shift. Interestingly, recent study [3] on uncertainty techniques in offline MBRL showed that the uncertainty measures proposed in the (non-private) literature actually correlate well with model error, more so than with distribution shift. Therefore, we believe that existing measures are appropriate for mitigating the worse reliability of the model under private training, since they will adequately capture the increased model error. We further point out that the impact of private model training also justifies increasing the reward penalty coefficient $\lambda$ to be more conservative, and a moderate increase in $\lambda$ was indeed beneficial in our preliminary experiments.
>
> [3] Lu et al., Revisiting design choices in offline model based reinforcement learning.

---

> ### Author Response · Authors · 2024-11-18
> **Rebuttal 3/3**
>
> We hope the discussions above have adequately addressed your concerns and answered your questions. Is there any additional information that may lead you to increase your score?

---

> > ### Comment · Reviewer_h3Xi · 2024-11-21
> >
> > Thank you for your feedback. I understand your points, but I believe your experiments are still insufficiently thorough. I will elaborate on my reasoning in the following points:
> >
> >  1. I think you should add an additional column in the main experiment table to show how  z  is configured, and then map it to the corresponding  \epsilon  values. Otherwise, it may give the impression that your experiments directly control  \epsilon , which could lead to misunderstandings.
> >  2. You mentioned that different benchmarks use different clipping methods and different uncertainty estimation techniques. I believe these choices should be presented as part of an ablation study rather than mixing them into the main experiment table. This approach would help readers intuitively understand why different methods are used for different benchmarks. The ablation study could also explain the reasoning behind adopting specific methods for each benchmark.
> >  3. Your current experiments are relatively basic and straightforward. I suggest including additional results on RL tasks that are of broader interest, such as performance on navigation tasks like AntMaze.
> >  4. I noticed that your supplementary experiments discuss a comparison with the work of Qiao & Wang (2023a). While I understand that their work is limited to linear MDPs and cannot handle more complex experimental scenarios, this section lacks a sufficiently thorough theoretical explanation of the specific implementation differences between the two frameworks to highlight the importance of your work. Additionally, from an experimental perspective, you should include a set of experiments that directly compare the privacy-utility trade-offs of both methods on the same benchmark. Relying solely on formulaic conversions and emphasizing that your framework can handle more complex benchmarks is insufficient.
> >  5. While I understand that the standard evaluation protocol is to assess privacy levels using  \epsilon , I would still like to see a demonstration experiment to clarify this point. For example, in the CartPole experiment, could you supplement your results by attacking policies with different privacy levels under equivalent conditions to demonstrate significant differences in the privacy levels of the two policies?
> >
> > If you can address and well explain the above points, I would consider increasing your score.

---

> > > ### Author Response · Authors · 2024-11-22
> > >
> > > Thank you for providing new suggestions and for elaborating your concerns over our experiments. We hope the following will adequately address your points.
> > >
> > > **Concerning your two first points**, we understand how your suggestions could help the understanding of the reader and will promptly take them into account in a new revision of our paper.
> > >
> > > **We would like to address the following points individually.**
> > >
> > > **3.** We emphasize that we are addressing the new task of learning DP agents in infinite-horizon discounted MDPs using general function approximation. We chose standard multi-dimensional, continuous tasks that cannot be represented in the episodic setting and are typically handled in the $\gamma$-discounted setting, and we focused on obtaining good results on these tasks to demonstrate the potential of our method. This is a major contribution as such tasks had not been tackled in the DP RL literature before. We agree that a future goal should be to deploy this approach to higher-dimensional tasks like AntMaze,  however we face several limitations that we clearly mention in the paper. In particular, a limiting aspect is the size of the dataset. As we discuss in Section 5 and further develop in section L of the appendix, the privacy-performance trade-off of our algorithm is greatly improved by using larger datasets, because of how the privacy guarantee relies on amplification by sup-sampling, and we argue that existing offline benchmark are not well adapted to study privacy. We therefore produce our own offline benchmarks with larger datasets. While it is unrealistic for us to collect, store and access datasets with millions of users because of computational constraints (especially as the task dimension grows), some applications (*e.g.*, in NLP) naturally induce such large volumes of data. To scale towards state-action spaces of higher dimension, future research directions involve 1) limiting the number of real trajectories accessed at each training epoch to enhance privacy amplification by sub-sampling for a fixed dataset size and 2) mitigating the impact of the task dimension on model perturbation. We believe a promising line of work for 1) is to augment each trajectory with data perturbation to generate more training data from fewer real examples. This would require linking gradient perturbation to the stochasticity of the input, which is an orthogonal line of research and an important open question in optimization. Concerning 2), we think an interesting direction would be to learn compact representations of high-dimensional inputs and perform planning directly in the latent space, as studied in [1].
> > >
> > > [1] Jiang et al., Efficient Planning in a Compact Latent Action Space.

---

> > > ### Author Response · Authors · 2024-11-22
> > >
> > > **4.** We agree that this comparison can benefit from a more thorough explanation about the differences between the two frameworks and will promptly incorporate this analysis in our paper. To summarize the key similarities and differences:
> > > - Both approaches are model-based.
> > > - PriMORL works for general function approximation, while DP-VAPVI from Qiao & Wang (2023a) relies on linear function approximation with known features.
> > > - We use and privatize a global model (that does not depend on the step $h$ ) for the environment that takes as input a pair $(s,a)$ and output $(s^\prime, r)$ . Qiao & Wang (2023a), on the other hand, learns one model (and one policy) per step, keeping track of $5H$ statistics to model the environment, where $H$ is the task horizon. Q-values and policies are computed recursively from step $H$ to step $1$.
> > > - Our approach is design for continuous state and action spaces. While DP-VAPVI from Qiao & Wang (2023a) could theoretically handle continuous actions, the policy is computed with a maximization step $\text{arg}max_{\pi_h(\cdot \vert s)} \langle \hat{Q}_h(s,a), \pi_h(\cdot \vert s)\rangle$ is impractical to compute when $\mathcal{A}$ is not finite.
> > > - For DP-VAPVI, given a total privacy budget $\rho$ , the privacy budget $\rho_0 = \frac{\rho}{5H}$ for each of the $5H$ statistics depends on the horizon: the larger $H$ , the smaller $\rho_0$ , and then the more noise we have to add to each statistic.
> > >
> > > This comparison further highlights that our method and the ones from Qiao et Wang are designed for very distinct settings. We cannot efficiently implement DP-VAPVI on our benchmark, in particular because of the continuous action spaces and the fact that it explicitly relies on a finite, relatively small horizon $H$ (not only the number of statistics to maintain and privatize depends on $H$ , but the amount of noise needed to privatize each statistic also grows linearly with $H$). However, it would be feasible to adapt our algorithm to handle the two-state synthetic MDP proposed in Qiao et Wang (2023a) for a direct comparison. If you consider that such an experiment could be interesting for the ICLR community, we will make sure to provide it.
> > >
> > > Moreover, we would like to emphasize that our goal is not to improve over Qiao et Wang but to address a new range of problems. We therefore only indirectly to Qiao et Wang to highlight the current state of the DP RL literature and the kinds of privacy-performance trade-offs that have been achieved so far.
> > >
> > > **5.** We agree that using privacy attacks against our algorithm would provide a better understanding of the protection they provide, and we mention it as important direction for future work. However, not to mention the fact that these attacks are very costly to implement, we also point out there is currently no rigorous, standardized benchmark to assess privacy empirically. We believe the development of such a benchmark is important to better understand the protection needed for deep RL algorithms, and to close the gap between between actual privacy leakage and overly loose theoretical upper bounds. However, it is a highly complex task that would be the focus of an entirely separate work.
> > >
> > > This evaluation protocol is considered standard because an upper bound $\epsilon$ offers a strong theoretical guarantee that the actual privacy leakage will not exceed this value in the worst case. While several studies focus on designing membership inference attacks on a selection of algorithms, we are aware of no work has simultaneously developed a differential privacy (DP) method and attacked it, due to the task's implementation and computational complexity, as well as its lack of rigor. This is made even more difficult due to the lack of open-source implementations of attacks framework (especially from [2], which would be the most relevant to our work).
> > >
> > >  [2] Gomrokchi et al., Membership Inference Attacks Against Temporally Correlated Data in Deep Reinforcement Learning
> > >
> > > **We would be glad to provide further clarification on any aspects, and we will make sure to upload a revised version of the paper with the discussed points in the coming days.**

---

> > > > ### Comment · Reviewer_h3Xi · 2024-11-24
> > > >
> > > > Thank you for your response. I am happy to increase my score.

---

> > > > > ### Author Response · Authors · 2024-11-25
> > > > >
> > > > > Thank you for the constructive discussion and valuable insights that have allowed us to improve the quality of our work.

---

### Official Review · Reviewer_pAHS · 2024-11-03

**Soundness:** 2
**Presentation:** 3
**Contribution:** 3
**Rating:** 6
**Confidence:** 2

**Summary:**

This paper studied differentially private model-based offline reinforcement learning (RL). The paper proposed a new algorithm that provide differential privacy for training ensemble models. Besides theoretical guarantees, the experiments also demonstrate the effectiveness of the proposed algorithm.

**Strengths:**

1. The proposed method that provides privacy guarantees for training ensemble models is novel.
2. The investigation of the problem is thorough.

**Weaknesses:**

1.  There exist several typos. For example, in line 274, 'it does entirely remove...' I assume it should be 'it does not entirely remove...' because increasing $N$ will degrade the model performance.
2. The major weakness is that there is a lack of a more explicit discussion on each term in the theoretical results, such as $\epsilon_p$ and $\epsilon_p^{DP}$. I would be curious about if these terms depend on privacy parameter or $N$, if so, what should be the approximate dependency.

**Questions:**

It seems that the number of ensembles $N$ plays a vital role in the results. This work has already reduce the dependency to $\sqrt{N}$ in the sensitivity. I am just curious is it possible to further reduce it in training ensemble models.

---

> ### Author Response · Authors · 2024-11-18
> **Rebuttal**
>
> Thank you for the positive feedback. In the following, we are pleased to provide you further insights about the points you raised.
>
> ``There exist several typos. For example, in line 274, 'it does entirely remove...' I assume it should be 'it does not entirely remove...' because increasing will degrade the model performance.``
>
> Thank you for pointing this out, it is indeed 'it does not entirely remove'. We apologize for the typos and did our best to correct them in the revised version of the paper.
>
> ``The major weakness is that there is a lack of a more explicit discussion on each term in the theoretical results, such as $\epsilon$ and $\epsilon^{DP}$. I would be curious about if these terms depend on privacy parameter or $N$, if so, what should be the approximate dependency.``
>
> We thank you for this feedback that made us rethink the role of the distribution shift in our results. After careful consideration, we ultimately determined that the distribution shift was not affected by the private model training. This is because the distribution shift term of the simulation lemma comes from a bound on $\vert V^\pi - V^{\pi^B}\vert$, the valuation gap between the policy $\pi$ and the data collection policy $\pi^B$, which does not depend on the model. Therefore, private training only impacts the model error term in proposition 4.3 and 4.4. We therefore have a better idea on how private training impacts the reliability of the model as a simulator in private offline MBRL. In the updated version of the paper, we have made modifications to Section 4.3 to account for these new findings.
>
> ``It seems that the number of ensembles plays a vital role in the results. This work has already reduce the dependency to $\sqrt{N}$ in the sensitivity. I am just curious is it possible to further reduce it in training ensemble models.``
>
> Thank you for the interesting question. The ensemble size indeed plays an important role in the trade-off between privacy and utility, and you are right to highlight the importance of controlling this parameter. When distributing the clipping norm across the $N$ models, the $\sqrt{N}$ factor directly comes from the fact the Gaussian mechanism uses the L2-sensitivity, suggesting that we could not further reduce the dependence in $N$ while using Gaussian noise for privacy. While other mechanisms could be considered, the Gaussian mechanism is the preferred choice for high-dimensional queries, especially because it can use the L2-sensitivity (a Laplace noise, on the other hand would linear scaling in the ensemble dimensions as it uses the L1-sensitivity) and allows tight computations of the privacy bound (*e.g.*, it enables the computations of the moments accountant). Therefore, we believe that a $\sqrt{N}$ dependence is the best we can get when using model ensembles.
>
> We hope these discussions have appropriately answered your questions. Is there any additional information that could lead you to increase your score?

---

> ### Author Response · Authors · 2024-11-22
>
> Dear Reviewer,
>
> Have you had the opportunity to review the discussions above and the revisions to the paper? If you have any further questions or concerns regarding the points you raised, we would be happy to discuss them.

---

> > ### Comment · Reviewer_pAHS · 2024-11-24
> >
> > Thanks for the reply. I have no further question and decide to retain my score.

---

> > > ### Author Response · Authors · 2024-11-25
> > >
> > > Thank you again for your time and positive feedback.

---

### Official Review · Reviewer_bCjw · 2024-11-12

**Soundness:** 3
**Presentation:** 3
**Contribution:** 2
**Rating:** 5
**Confidence:** 4

**Summary:**

This work extends model-based offline reinforcement learning, namely MORL, to its differentially private variant, namely PriMORL. The work intends to guarantee trajectory level DP, which means it treats two datasets that differ in at most one (entire) trajectory as neighboring datasets and asks the algorithm to output indistinguishably between them.

The differential privacy is achieved by model privacy, i.e. if the model is learned in a DP way, then by post processing the algorithm is also DP. This though comes with the limitation that the learning algorithm can no longer access the dataset once the model is obtained from the previous phase. To achieve DP model learning, it randomly draws a subset of trajectories and uses this batch of data to estimate a gradient. A clip is then applied to the gradient which bounds the sensitivity of the gradient. The work discusses different clipping techniques which fine tune the clipping threshold. The DP guarantee is then proved by moment accountant.

Once a model is obtained the policy is trained through pessimistic private MDP, which is by the intuition that being aware of the model uncertainty requires pessimisty. This intuition inspires the authors to run soft actor-critic on the  pessimistic variant of MDP, where the reward is reduced by the uncertainty level. This to some extent mitigates the cost of not being able to access the data once the model training is complete.

Many experiments are provided on tasks like pendulum, cartpole, and halfcheetah. The authors didn't seem to compare their algorithms with baseline methods, while there seems to be many. Though, the performance of the proposed algorithm is reasonably good on its own.

**Strengths:**

As a summary, this work investigates a natural setting and models the problem in a reasonable way. The work guarantees DP through model learning and post processing, which is intuitive. The performance of the algorithm seems decent.

**Weaknesses:**

1. Model/Techniques are not exciting.
2. No baseline comparison.
3. Limited testbeds

**Questions:**

N/A

---

> ### Author Response · Authors · 2024-11-18
> **Rebuttal 1/3**
>
> Thank you for your feedback.
>
> First, we would like to question the relevance of the argument about the "excitement" that a paper might generate. We do not understand why such an argument is present in a review, since the quality of the paper is normally assessed based on the rigor and relevance of the model and techniques developed. Here, we address a problem with many practical implications that has not currently been addressed in the literature: the possibility of doing differentially private deep RL based on offline data. We 1) explain why this problem is important and 2) fill in the literature by providing a practical solution achieving good performance on a standard (non DP) benchmark.  Once again, we are sorry if the reviewer finds the reading unexciting but we would like this work to be judged by the standard criteria: impact and scientific rigor.
>
> It has been shown that RL is not immune to privacy attacks, especially membership inference attacks where malicious adversaries could recover entire trajectories of the training dataset (see, e.g., [1]), and which could be prevented with differential privacy. Currently, the differentially private RL literature is limited to tabular and linear MDPs, and proposed methods do not scale to standard RL benchmark tasks. Given the well-founded concerns about data leakage in the case of RL applied to real-world situations, the literature on DP RL cannot be limited to theoretical studies (although these are useful) but must progress towards the deployment of algorithms in complex environments. To achieve this, deep RL solutions are needed. In this work, we are the first to address differentially private RL in the infinite-horizon discounted setting, without making limiting assumptions on the MDP. Furthermore, this work is the first in this area to deal with general function approximation, and to introduce differentially private deep RL algorithms evaluated on standard tasks beyond numerical simulations. Therefore, this work fills an important gap in the current DP RL literature and is impactful as a first step towards deploying private RL agents in complex environments. We thus believe that this work is highly relevant to the community.

---

> ### Author Response · Authors · 2024-11-18
> **Rebuttal 2/3**
>
> **Regarding the limitations of our experimental results and the lack of baseline, we would like to provide several clarifications.**
>
> Contrary to the above claim, we do perform baseline comparisons. As can be read in Section 5, we compare our algorithms to MOPO, a non-private baseline that is close to our method. Comparing a private method against a close non-private baseline is standard practice in the differentially private literature (see for instance [2], a concurrent work using a similar evaluation protocol).
>
> Moreover, to the best of our knowledge, the only work proposing DP algorithms in the offline RL setting is [2]. This work is limited to tabular and linear episodic MDPs, is only evaluated on a small synthetic environment (two state, discrete action space and fixed horizon $H=20$), and cannot scale to the problems we consider in our work, preventing direct comparison on the same benchmark.
>
> However, we point out that are our experimental results are overall much stronger than previous work in differentially private RL, where empirical evaluation is limited to numerical simulations in small environments (if there is any). By contrast, our work is the first to address standard control benchmarks with continuous state and action spaces using differentially private RL.
>
> Especially, our empirical results are stronger than [2]. Our method is indeed capable of achieving similar privacy-utility trade-offs than methods from [2], but on much more complex environments. We updated the paper by providing a clearer comparison with [2] (see experimental sections and Section F of the appendix), and provide details below.
>
> [2] do not mention explicitly the privacy budgets $\epsilon$, but instead mention the zero-concentrated differential privacy (z-CDP) parameter $\rho$. For clarity and fair comparison, we convert the z-CDP guarantee into a DP guarantee, using Proposition 1.3 from [3]: if a mechanism is $\rho$-z-CDP, then for any $\delta > 0$ it is $(\epsilon, \delta)$-DP, with $\epsilon = \rho + 2 \sqrt{\rho \log(1 / \delta)}$. As they evaluate their algorithms for a dataset size up to 1000, we consider two values of $\delta \in \{1/100, 1/1000\}$. The table below shows the results for the various parameters $\rho$ mentioned in Figure 1 from [1].
>
> | $\rho$ | $\epsilon$ for $\delta=10^{-1}$ | $\epsilon$ for $\delta=10^{-3}$ |
> |--------|----------------------------------|----------------------------------|
> | 25     | 40.2                             | 51.3                             |
> | 5      | 11.8                             | 16.8                             |
> | 1      | 4.0                              | 6.26                             |
> | 0.1    | 1.1                              | 1.8                              |
>
> Therefore, [1] also considers the low privacy regime with $\rho=25$ yielding $\epsilon$ close to 50, which is comparable to our low privacy variant. They indeed consider $\epsilon$ close to 1 with $\rho=0.1$, but the cost is a 2 to 3 times worse utility on this simple MDP. Other configurations proposed are closed in privacy budgets to what we consider in our paper. Overall, our work achieves comparable privacy-utility trade-offs than [1], but on significantly more complex tasks.
>
> [1] Gomrokchi et al. Membership Inference Attacks Against Temporally Correlated Data in Deep Reinforcement Learning.
>
> [2] Dan Qiao and Yu-Xiang Wang. Offline reinforcement learning with differential privacy.
>
> [3] Bun and Steinke, Concentrated Differential Privacy: Simplifications, Extensions, and Lower Bounds.

---

> ### Author Response · Authors · 2024-11-18
> **Rebuttal 3/3**
>
> After considering these discussions and following the revisions made to our paper, is there any additional information that may make you increase your score?

---

> ### Author Response · Authors · 2024-11-22
>
> Dear Reviewer,
>
> We wanted to follow up and see if you have had a chance to review the revisions and the discussions mentioned above. If you have any additional questions or concerns regarding the issues you raised, we would be glad to address them.

---

> ### Comment · Reviewer_bCjw · 2024-11-24
>
> Thank you for providing a rebuttal. By saying "not excited" I mean 1) there's a lack of novelty. This work combines model learning, DP, SAC in the offline RL setting; 2) I do not see real applications that we are incentives to use offline RL facing a privacy setting. I apologize if this term caused some confusion. I hope this clarifies that we are on the same set of scientific standards.
>
> By the same argument, I would expect the experiment to work on much more complicated tasks that showcase your method outperforms other private learning methods (within and beyond offline RL). This is why I evaluate the experiments as limited. I understand that the authors would like to compare the manuscript with some other works published at a major ML conference. Unfortunately I did not use this method to evaluate the manuscript.
>
> By these reasons the rebuttal does not change my evaluation, at this moment.

---

> > ### Author Response · Authors · 2024-11-25
> >
> > Thank you, we appreciate the clarifications and will do our best to address your concerns.
> >
> > We believe that the study of offline RL in the private setting is well-motivated. Privacy threats in RL are well documented (see [1, 2, 3]): an adversary can use a trained policy to infer sensitive information about the training data (like the membership of a specific trajectory). It is therefore in our interest to implement data protection mechanisms in any situation where an RL agent is trained on personal data (we could even be legally required to do so by regulations such as GDPR in Europe). We can think of many such examples of real-world applications where RL agents are trained using sensitive data, including:
> > -Autonomous vehicles [4]: trained on a large number of trips that may disclose, for instance, locations and driving habits
> > - Healthcare: RL agents for personalized treatment recommendation [5] are trained on patients' health and treatment history
> > - Recommmendation engines [6]: trained on browsing journeys that can reveal user preferences on various sensitive topics, e.g., politics
> > - Personalized Finance: RL-based automated investment managers [7], based on users' history of interaction with financial markets
> > - Banking and Insurance: RL agents can be trained to assess credit risk [8] or modulating insurance policy offers [9]
> >
> > We also note that in industry, where interacting with systems online is difficult, costly, or even dangerous, agents are often trained offline, which further increases our interest for private offline RL.
> >
> > *References*:
> >
> > [1] Gomrokchi et al., Membership Inference Attacks Against Temporally Correlated Data in Deep Reinforcement Learning
> >
> > [2] Pan et al., How You Act Tells a Lot: Privacy-Leaking Attack on Deep Reinforcement Learning
> >
> > [3] Prakash et al., How Private Is Your RL Policy? An Inverse RL Based Analysis Framework
> >
> > [4] Ravi Kiran et al., Deep Reinforcement Learning for Autonomous Driving: A Survey
> >
> > [5] Liu et al., Deep reinforcement learning for personalized treatment recommendation
> >
> > [6] Mehdi-Afsar et al., Reinforcement learning based recommender systems: A survey
> >
> > [7] Hambly et al., Recent Advances in Reinforcement Learning in Finance
> >
> > [8] Paul et al., An Automatic Deep Reinforcement Learning Based Credit Scoring Model using Deep-Q Network for Classification of Customer Credit Requests
> >
> > [9] James Young et al., Reinforcement Learning applied to Insurance Portfolio Pursuit

---

> > ### Author Response · Authors · 2024-11-25
> >
> > We understand that the current limitations of our method prevent deployment on more complex tasks closer to real-world situations, but we respectfully emphasize the importance of re-contextualizing our work within the current state of the literature. Despite the numerous potential applications highlighted above, there is only a limited amount of work in differentially private RL, and even fewer in the offline setting. These works are limited to theoretical aspects, which, although very useful, do not offer solutions that could scale to the kind of problems described above. This is in contrast to other areas of ML such as supervised learning, where private deep learning methods (e.g., DP-SGD) have been widely studied for years and can already be deployed at scale. As RL faces similar privacy issues as supervised learning, we thus consider that a paradigm shift is needed, where we build towards private RL algorithms that can be deployed at scale. We need deep RL agents with provable privacy guarantees, and our work is the first step in this direction.
> >
> > Regarding our method itself, we indeed build on techniques from the offline and model-based RL literature, but we respectfully disagree that our method is a straightforward combination of all of these. Although the approach can seem natural, no work has previously proposed and implemented similar techniques for the novel problem of private deep RL. Moreover, we have addressed various challenges specific to our private setting. As we could not just use DP-SGD to train the model, we adapted training techniques from the federated literature to enable model training with trajectory-level privacy, and carefully controlled the privacy budget for model ensembles. We also carefully proved privacy guarantees for the end policy by limiting online interactions during policy optimization. We furthermore provided theoretical insights on the cost of private gradient-descent training in the model-based setting. Finally, our empirical results set a new standard for differentially private deep RL, encouraging the development of this field. If our approach cannot yet scale to larger problems, we acknowledge its limitations and identify areas of improvement for future work, such as mitigating the impact of the task dimension on model perturbation (for instance using planning in latent spaces) and limiting the number of real trajectories accessed at each training epoch (for instance using data augmentation).
> >
> > We are also a bit confused regarding the comparison to other private learning methods since we propose the first private deep offline RL algorithm for general MDPs. As stated earlier, the only comparable methods would be the algorithms from Qiao et al. (2023a), which we compare to indirectly as they do not apply to the kind of problems we address in this work. Could you be more specific about the kind of comparisons you have in mind?
> >
> > We hope this discussion has adequately addressed your concerns, and thank you again for your time and valuable feedback. If you have any additional question or remark at this point, we would be happy to engage in further discussion.

---

> ### Comment · Reviewer_bCjw · 2024-11-27
>
> Thank you for the further discussion. I basically agree with the arguments you have provided. I believe such arguments alone do not justify sufficient novelty and sufficient relevance, though. Therefore my evaluation remains the same. I apologize for not being able to increase the score.

---

> > ### Author Response · Authors · 2024-11-28
> >
> > Thank you for taking the time to engage in these discussions.

---

### Author Response · Authors · 2024-11-18
**Paper Revision**

We thank the reviewers for their constructive feedback. We have incorporated all the comments in a new version of our paper, which we have just uploaded. We hope that these modifications, along with the discussions, adequately address your concerns and answer your questions.

---

### Author Response · Authors · 2024-11-28
**New Paper Revision and General Comment (1/2)**

We would like to thank all the reviewers for their feedback and for engaging in valuable discussions during the rebuttal period, providing us with meaningful insights. Efforts made to address the reviewers' concerns enabled us to make impactful improvements to our paper. We have uploaded a revision of our work in which we highlight, with blue text, the significant modifications and additions we made following this rebuttal.

We would like once again to emphasize the relevance, impact, and contributions of our work. In particular, we highlight why the study of private offline RL is well-motivated, re-contextualize our work within the current private RL literature, offer evidence of the novelty and impact of our work, and emphasize the challenges of the setting and our technical contributions.

In this paper, we address offline deep reinforcement learning with differential privacy (DP) guarantees. We use the well-motivated notion of trajectory-level privacy. We propose a model-based approach, PriMORL, and assess it empirically on several continuous control tasks.

**Motivations for the study of private offline RL**
- There are numerous current or potential applications of RL in risk-sensitive scenarios, including recommendation engines, healthcare, personalized finance, and autonomous vehicles. In all these scenarios, RL agents are trained on sensitive, personal user data.
- Privacy threats in RL are well documented, including powerful membership inference attacks. Studies suggest RL is no more immune to privacy leakage than supervised learning or any other field.
- Offline RL is particularly relevant in practical or industrial applications, where online interaction with the environment is often impractical, costly, and/or hazardous.

**Current state of the private RL literature**
- Existing DP approaches in both online and offline RL are predominantly theoretical and have limited practical impact. Specifically, they are limited to episodic tabular and linear MDPs, and the few experiments are restricted to numerical simulations.
- Existing methods cannot intrinsically scale to problems typically encountered in deep RL.
- There is currently no DP RL method for general MDPs in the $\gamma$-discounted MDP setting. More generally, there is currently no DP method for deep RL.

**Why this work is novel and impactful**
- While reinforcement learning encounters the same privacy challenges as other areas of ML, no existing work has proposed a private RL method that matches the versatility, scalability, and empirical effectiveness of \textsc{DP-SGD} for supervised learning. We believe this observation justifies a shift from predominantly theoretical work to practical deep RL approaches with the ability to scale to complex problems.
- We propose the first deep RL approach with formal differential privacy guarantees. It works in general, continuous MDPs in the $\gamma$-discounted infinite-horizon setting.
- For the first time in the DP RL literature, we tackle standard control tasks with deep function approximations. Experiments show that our method can learn deep DP RL agents with limited performance cost compared to the non-private baseline. These results set a new standard for practical DP RL.
- Despite its current limitations, we identify promising directions to scale this approach towards high-dimensional deep RL problems. Overall, this work takes a first step into bridging towards the deployment of private RL agents in practical risk-sensitive applications.

---

> ### Author Response · Authors · 2024-11-28
> **New Paper Revision and General Comment (2/2)**
>
> **On the challenges and technical contributions of this work**
> - While the notion of trajectory-level privacy is well-motivated, this is not straightforward to achieve in deep RL, since standard approaches like \textsc{DP-SGD} deal with privacy at the example-level. We identify that trajectory-level privacy requires partitioning the data by trajectories before computing and clipping trajectory-level updates. We build on prior work that tackles user-level privacy in the fields of federated learning and NLP. We adapt these ideas to our setting and propose a trajectory-level DP training method based on a Poisson sampling scheme to learn the dynamics model.
> - The standard approach to handle model uncertainty is using bootstrap ensembles to compute uncertainty estimates. This raises additional challenges in the private setting, where the size of the ensemble impacts the privacy budget. We mitigate this issue by distributing the clipping factor across all models.
> - We provide a theoretical analysis of how private training influences model reliability and its impact on the policy optimization process.
> - We prove formal theoretical guarantees for the end policy by restricting further interactions with the system during policy optimization.
> - We identify that current offline RL benchmarks are unsuitable for studying privacy and advocate for the use of larger datasets.
> - We obtain meaningful experimental results on previously unaddressed benchmarks, paving the way for future work in private deep RL.
> - We release our code to ensure reproducibility and encourage the development of private deep RL.
>
> During this rebuttal period, we have carefully addressed the concerns and questions of all the reviewers, and these have all been taken into account in the revised version of the paper. In particular, we did our best to further highlight the motivation and contributions of our work, addressing the concerns of reviewers bCjw, h3Xi, and wk8A. We also propose promising research directions to scale towards higher-dimensional problems, following discussions with reviewers bCjw and h3Xi. We provided a detailed comparison with the work of Qiao \& Wang (2023a), the closest to our work, to answer concerns from reviewers bCjw and wk8A. We are also more explicit on the role of $\epsilon$ and the computations of the moments accountant, following a question from reviewer pAHS. We thank again the reviewers for their constructive feedback, which allowed us to improve our work in these directions.

---

### Meta-Review · Area_Chair_Dqgo · 2024-12-22

**Metareview:**

This paper studies the problem of differentially private learning in the context of offline RL. The primary concern with the paper is on (1) novelty: extending DP to offline RL is straight-forward conceptually, especially using a model-based method. One can simply apply a DP supervised learning method to learn the model and then do planning in the learned model as one would expect. The trajectory-level DP seems to be an non-substantial deviation; (2) Lack of convincing experimental evaluation: While the authors claim that the method in this paper is more scalable than prior work, that is not true. Continuous control task with low-dimensional feature is considered as simple as tabular/linear MDPs. If the authors wish to claim practical relevance for the proposed algorithm, they must demonstrate its performance on benchmarks with visual input.

**Additional Comments On Reviewer Discussion:**

NA

---

### Decision · Program_Chairs · 2025-01-22

Reject